# Mapping the gene regulatory landscape of archaic hominin introgression in modern Papuans

Maddy Comerford[1,2], Davide M. Vespasiani[2¤a], Navya Shukla[1,2], Laura E. Cook[2¤b], Danat Yermakovich[3,4], Michael Dannemann [3], Matthew Leavesley [5,6,7], Christopher Kinipi[8], François-Xavier Ricaut [4], Nicolas Brucato[4], Murray P. Cox[9,10], Irene Gallego Romero [1,2,11,12] *

**1** Human Genomics and Evolution, St Vincent's Institute of Medical Research, Melbourne, Australia, **2** School of Biosciences, University of Melbourne, Melbourne, Australia, **3** Center for Genomics, Evolution and Medicine, Institute of Genomics, University of Tartu, Tartu, Estonia, **4** Centre de Recherche sur la Biodiversité et l'Environnement (CRBE), Université de Toulouse, CNRS, IRD, Toulouse INP, Université Toulouse 3 – Paul Sabatier (UT3), Toulouse, France, **5** Strand of Anthropology, Sociology and Archaeology, School of Humanities and Social Sciences, University of Papua New Guinea, Port Moresby, Papua New Guinea, **6** ARC Centre of Excellence for Australian Biodiversity and Heritage, College of Arts, Society and Education, James Cook University, Cairns, Australia, **7** ARC Centre of Excellence for Indigenous and Environmental Histories and Futures, College of Arts, Society and Education, James Cook University, Cairns, Australia, **8** Health Services, University of Papua New Guinea, Port Moresby, Papua New Guinea, **9** Department of Statistics, University of Auckland, Auckland, New Zealand, **10** College of Sciences, Massey University, Palmerston North, New Zealand, **11** Faculty of Health Sciences, Australian Catholic University, Melbourne, Australia, **12** School of Medicine, University of Melbourne, Melbourne, Australia

¤a Current Address: Genetics and Gene Regulation Division, The Walter and Eliza Hall Institute of Medical Research, Parkville, Australia.
¤b Current Address: Lawrence Berkeley National Laboratory, Berkeley, California, United States.
* irene.gallego@svi.edu.au

## Abstract

Interbreeding between anatomically modern humans and archaic hominins has contributed to the genomes of present-day human populations. However, our understanding of the specific gene regulatory consequences of Neanderthal, and particularly, Denisovan introgression is limited. Here, we used a massively parallel reporter assay to investigate the regulatory effects of 25,869 high-confidence introgressed SNPs segregating in present-day individuals of Papuan genetic ancestry in immune cell types. Overall, 8.22% of Denisovan and 8.58% of Neanderthal sequences showed active regulatory activity, and 9.22% of these displayed differential activity between archaic and modern alleles. We found no association between introgressed allele frequency on activity regardless of introgression source, but introgressed Denisovan alleles at higher frequencies were less likely to be differentially active than expected, suggesting introgression is under some degree of selective constraint. Both activity and differentially activity were associated with distance to the nearest transcription start site, while differential activity was additionally associated with differential transcription factor binding. Genes predicted to be regulated by differentially

**Data availability statement:** Raw sequencing reads have been deposited to ENA under accession number PRJEB88773. Code for all analyses is available here: https://gitlab.svi.edu.au/igr-lab/png_introgression.

**Funding:** This work was supported by an award from the Leakey Foundation and by Australian Research Council Discovery Project DP200101552, both to IGR, by the Marsden Fund of the Royal Society of New Zealand (20-MAU-017) to MPC and IGR, by French National Research Agency (ANR) (grant PAPUAEVOL ANR-20CE12-0003-01 to FXR, by Estonian Research Council grant TK (TK214) to MD, and by the Commonwealth through an Australian Government Research Training Program Scholarship to MC. St Vincent's Institute acknowledges the infrastructure support it receives from the National Health and Medical Research Council Independent Research Institutes Infrastructure Support Program and from the Victorian Government through its Operational Infrastructure Support Program. The funders had no role in study design, data collection and analysis, decision to publish, or preparation of the manuscript.

**Competing interests:** The authors have declared that no competing interests exist.

active sequences included *IFIH1* and *TNFAIP3*, key immune genes and known examples of archaic introgression. Overall, this work provides experimental validation of regulatory activity for thousands of archaic variants in populations with the highest levels of Denisovan ancestry worldwide, revealing how human evolutionary history actively shapes present-day genetic diversity and immune function.

## Author summary

Modern humans carry DNA inherited from extinct archaic hominins, the Neanderthals and Denisovans. Archaic DNA influences present-day traits, including immunity, but our understanding of the mechanisms of this process are limited. Here, we characterise the gene regulatory potential of Neanderthal and Denisovan variants present in Papuans, who have the highest levels of Denisovan ancestry globally. We identify variants capable of modulating gene expression, many of which are predicted to impact immune-related genes. Our data supports a role of Neanderthal and Denisovan DNA in the immune response of modern-day Papuans and provides examples of how human evolutionary history shapes present-day diversity. By applying a novel technology to answer evolutionary questions, we highlight how functional genomic approaches will be useful in driving our understanding of global genetic variation.

## Introduction

As modern humans left Africa and spread across the globe, they came into contact with multiple archaic human groups, two of which we have published genome sequences for: Denisovans [1,2] and Neanderthals [3–6]. Multiple interbreeding events between expanding anatomically modern humans (AMH) and archaic groups led to present-day humans carrying introgressed archaic DNA. Reflecting a complex population history, this introgression is not uniformly distributed across AMH populations today. Neanderthal introgression has contributed up to 2% to the genomes of individuals of non-African genetic ancestry [4], while Denisovan introgression accounts for an additional 3% of the genomes of individuals of Papuan or Near Oceanic genetic ancestry [1,2,7] and under 1% in individuals of South or East Asian genetic ancestry, and is virtually absent elsewhere in the world.

Additionally, archaic introgression is not distributed uniformly across the genome. While selection against introgressed DNA appears to have been widespread across the genome [8,9], we and others have previously shown a significant excess of introgressed DNA at or near genes involved in immune function [10–22], suggesting that introgression was not always disadvantageous. Particularly in the case of immune phenotypes, introgressed alleles would have been 'fine tuned' relative to those carried by expanding AMH, and better suited to countering the threats posed by local pathogens. Their retention would thus have conferred a selective advantage

to carriers [18]. Much of this advantage is thought to be mediated not by changes to protein coding genes (although there are notable exceptions, e.g., [23]) but primarily by driving fine-scale changes in gene expression levels. For example, Neanderthal sequences are overrepresented amongst steady-state and response eQTLs [11,24]. Additionally, introgressed Neanderthal sequences have been predicted to impact allele-specific gene expression levels [25] and to affect enhancers in a tissue-specific manner [26].

However, our understanding of the specific functional consequences of Denisovan introgression in present-day human populations remains limited. In comparison to Neanderthal introgression, Denisovan introgression is under-represented in genomic databases, because Denisovan DNA is not present in the genomes of European populations who make up the large majority of genomic databases [27], and much introgression falls in non-coding parts of the genome where function is harder to predict directly from sequence. While we previously showed that in the genomes of present-day Papuan individuals, both Neanderthal and Denisovan introgressed alleles are enriched within potential tissue-specific gene regulatory elements, and in particular within chromatin regions active in various immune cell types [10], we could not validate these predictions against actual gene expression data due to the near complete absence of individuals of Papuan genetic ancestry from gene expression datasets [28].

Massively parallel reporter assays (MPRAs) allow for functional testing of regulatory activity at scale, and are rapidly becoming part of the evolutionary and comparative genetics toolkit [29]. In an MPRA, a synthesised library of short synthetic oligos predicted to regulate gene expression levels is tagged with unique barcodes and cloned in front of a minimal promoter and a reporter gene, such that the functional elements will drive transcription of the barcode and reporter gene [30]. In this manner, thousands, hundreds of thousands, or millions of sequences can be tested simultaneously, overcoming the low throughput of traditional reporter assays [31]. While initially applied to biomedical questions, MPRAs are increasingly being used to understand evolutionary questions (e.g., [32–35]). Particularly in light of the systematic under-representation of peoples of non-European genetic ancestry in public genetic and cellular resources, MPRAs provide a tractable and powerful way to examine the gene regulatory impact of archaic introgression in present-day Papuans.

Thus, to characterise the regulatory activity of archaic variants present in these individuals in immune cells, we have tested the activity of over 25,000 Denisovan and Neanderthal variants segregating in present-day Papuan populations at high frequencies. We identify Denisovan and Neanderthal alleles that robustly drive reporter gene expression, and by testing both the archaic and non-archaic allele of each variant, identify variants for which modern and introgressed alleles drive significantly different reporter gene expression. This work provides, for the first time, experimental insight into the regulatory activity of archaic variants segregating within Papuan populations and, more broadly, insights into the contribution of archaic introgression in shaping modern human genetic diversity.

## Results

### Identifying cis regulatory activity of introgressed sequences using an MPRA

To characterise the functional impact of archaic introgression on the gene regulatory landscape of modern-day Papuan populations, we tested the regulatory activity of 25,869 introgressed alleles and their modern human counterparts using an MPRA (Fig 1A). Starting with the set of introgressed SNPs identified in Papuan populations by Jacobs et al. [36], we followed the same approach as in Vespasiani et al. [10] and focused on those located in open chromatin, defined by DNAseI hypersensitive sites from [37,38] or Tn5 transposase accessible regions from [39], or in active chromatin states defined by the Roadmap Epigenome Project for at least one immune cell type (defined as all cells within HSC and B cells or blood and T cells in Kundaje et al. [37]; S1, S2 Figs). We did not require the annotations to be unique to immune cells; indeed, most annotations were shared across at least one immune and one non-immune cell type (S3 Fig). We also included introgressed SNPs predicted to disrupt transcription factor binding sites [10] (Fig 1B). Finally, we limited ourselves to variants with an introgressed allele frequency (IAF) $\geq$ 0.15 (S4 Fig), reasoning they were more likely to confer a selective advantage than those at lower frequencies. To quantify the activity of these variants we designed 200 bp

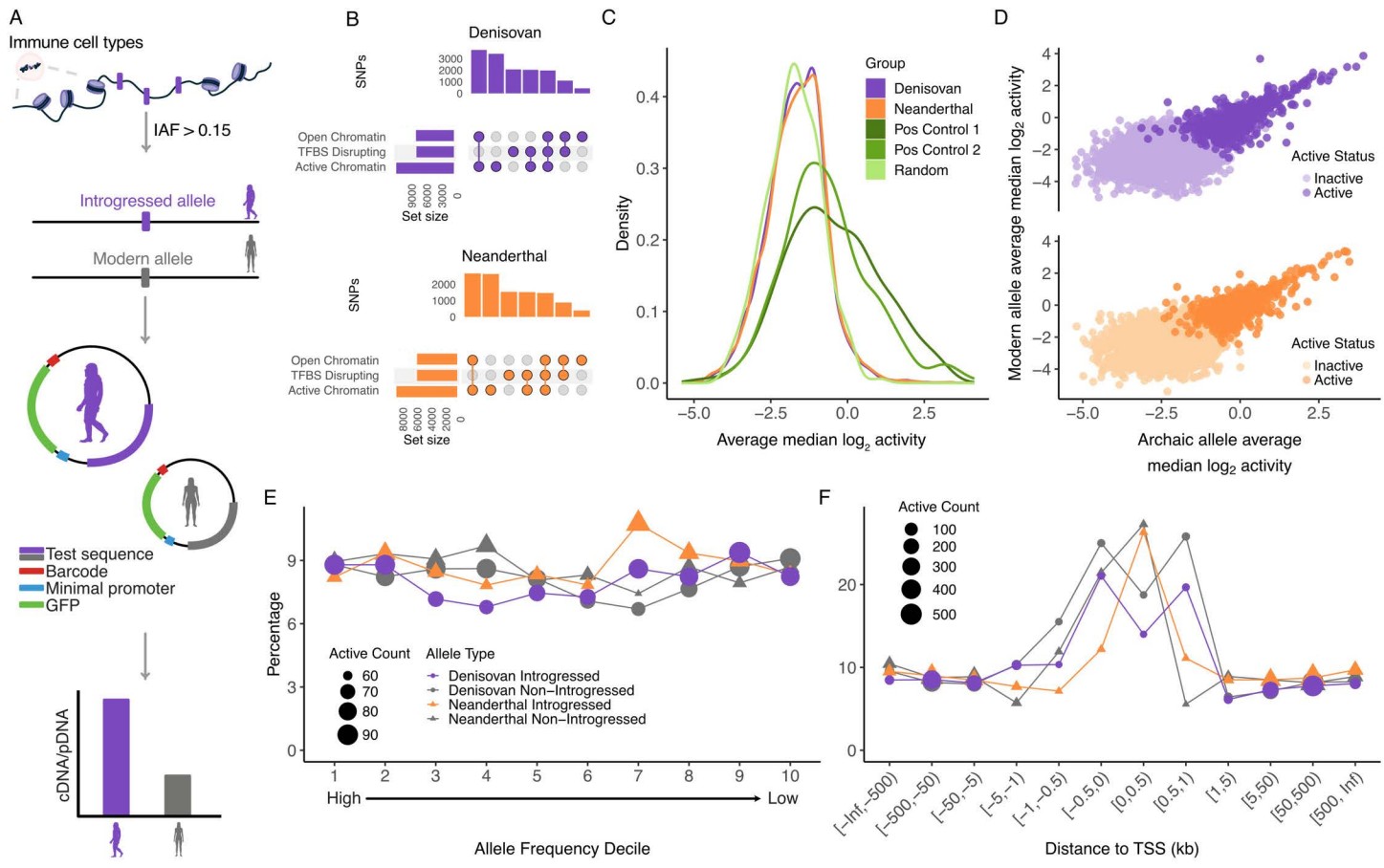

**Fig 1. Quantifying the regulatory activity of archaic SNPs with an MPRA. A.**) SNP selection and experimental overview. Human silhouettes are from [phylopic.org](phylopic.org) [42]; silhouette images are by **T.** Michael Keesey (*Homo neanderthalensis*) and Carlo De Rito (*Homo sapiens*). **B.**) Number of selected Denisovan (top) and Neanderthal (bottom) SNPs annotated across various functional annotations. **C.**) Distribution of activity for Denisovan, Neanderthal and control sequences. Positive control 1 and 2 sequences were previously found to be active by Tewhey et al. [40] and Arensbergen et al. [41], respectively. **D.**) Scatter plot of activity of the archaic allele and the modern allele for Denisovan (top) and Neanderthal (bottom) SNPs, with SNPs with at least one active allele highlighted. **E.**) Percentage of active alleles in allele frequency deciles. **F.**) Percentage of active alleles in binned distances to the transcription start site (TSS).

oligos containing 15 bp flanking adapter sequences surrounding 170 bp centred on the SNP of interest. For each allele, we included in our library an oligo containing the modern human allele, and an oligo containing the introgressed archaic allele. Where two or more SNPs were located within 170 bp of each other, which happened 15.39% of the time, we additionally designed haplotype-like oligos containing all possible allelic combinations of the SNPs. In total, our initial MPRA library contained 61,812 sequences across 25,869 high-confidence introgressed SNPs. This included 35,198 and 25,496 single and haplotype-like sequences testing 14,775 and 11,094 Denisovan and Neanderthal SNPs, respectively, which below we refer to as Denisovan and Neanderthal sequences. We also included 153 SNPs previously found to be active in a previous MPRA [40] and 196 SNPs active in HepG2 cells [41] to act as positive controls, and a set of 300 random scrambled sequences to act as negative controls. All tested sequences are available as S1 File.

After barcode addition and initial library quantification (see Methods), we transfected the library into two replicates each of three different lymphoblastoid cell lines (GM12878, a European LCL and PNG8 and PNG22, established from Papuan donors sampled in Port Moresby). We collected both RNA and plasmid DNA (pDNA), and sequenced barcode counts in the cDNA and

pDNA fractions, to an average depth of 247 million reads per library. We were unable to sequence cDNA from PNG8 replicate 1 due to low RNA recovery, so this replicate was discarded from downstream analyses. After filtering (see Methods), across replicates we captured a total of 44,144 sequences for activity testing, out of 59,954 (73.63%) of the sequences captured after library quantification. Measured sequences were associated with 48 barcodes on average (SD = 40, range = 10–562; S5 Fig). Correlations between the pDNA counts across replicates (Spearman's $\rho$ = 0.90 to 0.91, p < 2.2 × 10$^{-16}$), were higher than that of the correlations between the cDNA counts (Spearman's $\rho$ = 0.59 to 0.70, p < 2.2 × 10$^{-16}$). Sequence counts were moderately correlated between the cDNA and pDNA fractions within replicates (Spearman's rank correlation $\rho$ = 0.62 to 0.74, p < 2.2 × 10$^{-16}$), with a subset of sequences showing higher cDNA than pDNA counts (S6 Fig).

In total, 3,894 sequences (8.82% of measured sequences) showed consistent signs of regulatory activity in at least two replicates, defined as having a ratio of cDNA/pDNA counts across all barcodes that was significantly higher (FDR < 0.05) than the mean value for that replicate (see Methods). Positive control sequences showed higher activity than our archaic sequences and random scrambled sequences (Fig 1C), with 91 control sequences previously identified to be active in LCLs [40], and 154 control sequences previously identified to be active in HepG2 cells [41] labelled as active (47.89% and 38.4%, respectively, of sequences tested in these groups), validating our experimental approach. Considering modern and introgressed alleles together, at least one allele in 2,252 tested sequences was active (S2 File); this includes 2,059 alleles across 1,254 Denisovan sequences and 1,568 alleles across 998 Neanderthal sequences. Both alleles were deemed active in 571 of tested Denisovan sequences and 457 of Neanderthal ones. Overall, 8.22% and 8.58% of tested Denisovan and Neanderthal sequences respectively displayed some activity above background levels (Fig 1D). Of these, 1,699 Denisovan (845 introgressed alleles, 854 AMH alleles) and 1,376 Neanderthal sequences (692 introgressed alleles, 684 AMH alleles) contained only a single allele, while the remainder contained multiple alleles in a haplotype construct (discussed further below). Finally, 22 of 249 measured random sequences (8.84%) were also deemed active, but these were less likely to be active across all replicates than test sequences or positive controls (S7 Fig). Activity ratios were moderately correlated across replicates and, as expected, were substantially higher when considering summed activity (S8 FigA). Activity of the positive control sequences showed the highest correlation of all sequence types (S8 FigB-E, S9 Fig).

## Active introgressed alleles sequences display evidence of selective constraint

We then sought to characterise properties of active sequences to better understand what genomic features drive activity, as well as the features of functional archaic introgression. Approximately a quarter of active sequences were common to all five replicates (S7 Fig). When taking ancestry into account, we observed 145 Neanderthal sequences (out of 1,568 active Neanderthal sequences) and 128 Denisovan sequences (out of 2,059 active Denisovan sequences) that were active only in the GM12878 replicates, a significant excess of Neanderthal sequences (two sample proportion test p = 0.0007). Conversely, more Denisovan sequences were active only in the three Papuan replicates (91 out of 2,059) than Neanderthal sequences (80 out of 1,568) but the difference in this case was not significant (two sample proportion test p = 0.378). Furthermore, we did not see a bias towards either the modern or introgressed allele being active more often; approximately half of Denisovan and Neanderthal active sequences (49.7% and 50.3% respectively) contained the introgressed allele. Activity of the introgressed and modern alleles was clearly correlated (Fig 1D; Denisovan Pearson's r = 0.667, p < 0.001; Neanderthal Pearson's r = 0.669, p < 0.001). Activity was associated with GC content, with active sequences showing higher GC content than inactive sequences (S10 Fig; Denisovan Wilcoxon p = 3.21 × 10$^{-21}$, Neanderthal Wilcoxon p = 3.69 × 10$^{-22}$). In addition, active sequences occurred in all regulatory annotation classes (S11 FigA and B). There was no significant association between immune cell specific annotations and activity, but active alleles were enriched in open chromatin annotations shared across cell types and depleted from closed chromatin (S12 Fig).

We then asked whether allele frequency (AF) influences activity, as AF may be reflective of patterns of selection acting on functional archaic introgression. Because Neanderthal and Denisovan alleles segregating in genetically Papuan

individuals have different population histories, we examined whether activity patterns change by decreasing AF decile for both introgressed and AMH alleles (Fig 1E, S13 FigA), finding no significant association between AF decile and the percentage of active sequences within a decile (ANOVA p = 0.607). For each decile, we tested whether the percentage of active sequences in the decile is different to the overall percentage of active sequences (separately for Denisovan and Neanderthal sequences), but did not identify any significant differences (S14 FigA). For Denisovan sequences, the first decile contains SNPs segregating at IAF ≥ 0.31, and at IAF ≥ 0.45 for Neanderthal sequences; the lack of an association between IAF and activity in general does suggest that introgression can rise to high frequencies even when functional.

However, we observed a significant association between distance to the nearest annotated transcription start site (TSS) and the likelihood of a sequence being active (Fig 1F), with the percentage of active sequences being approximately 10% regardless of introgression source or allelic state when located > 1kb from the nearest TSS (S14 FigB). Sequences located ≤ 1kb from a TSS were much more likely to be active ($\chi^2$ p = $2.2 \times 10^{-16}$), likely reflective of general genome architecture, as well of the limitations of the MPRA system to capture small effects typically associated with distant enhancers. When focusing on SNPs within 1kb of a TSS, we observe no effect of allelic state on the likelihood of activity for either Denisovan (logistic regression p = 0.149) or Neanderthal (logistic regression p = 0.673) sequences.

## Identifying differentially active introgression

Next, we tested for differential activity between the two alleles of active introgressed sequences. Throughout this section, we considered all 1,951 tested single-variant sequences with at least one active allele, and compared active alleles to their inactive counterparts when relevant. Using mpralm [43] we identified 180 sequences (95 Denisovan and 85 Neanderthal) with differential activity between the introgressed and non-introgressed alleles at an FDR threshold of 0.05 (9.22% of archaic SNPs tested for differential activity; (Fig 2A, 2B, S3 File)). Amongst these, the introgressed allele drives higher activity in 44 and 55 Denisovan and Neanderthal sequences respectively (46.3% and 64.7%); and 27 sequences (12 Denisovan, 15 Neanderthal) exceed $\log_2$ fold change values of 1. Of the differentially active Denisovan sequences, both the introgressed and non-introgressed allele were active in 54 cases, while the remaining 41 showed activity in one allele only. Of the 85 differentially active Neanderthal sequences, 35 showed activity in both alleles, with 50 showing activity in one allele only.

As above, we asked whether we could identify overall trends across differentially active alleles. Constraint on function at high IAF is clear when considering patterns of differentially activity across deciles (Fig 2C, S15 FigA). Denisovan and Neanderthal differentially active sequences display different patterns, however. The proportion of differentially active sequences is correlated with IAF decile for Denisovan sequences (Pearson's r = 0.710, p = 0.021), and increases from 3.7% in the highest IAF decile to an average of 10.4% for alleles in the bottom 60% of the Denisovan sequences. IAF decile is less strongly associated with differential activity for introgressed Neanderthal sequences (Pearson's r = -0.027, p = 0.941). However, the general paucity of introgressed alleles segregating at high allele frequencies (S4 Fig) means that the first IAF decile spans a large range of IAF values and obscures more fine-grained trends (S13 FigB). In line with this, when testing for enrichment of differentially active sequences at varying IAF thresholds, we observe no enrichment or depletion for Neanderthal SNPs at any threshold, while we observe a significant deficit of Denisovan differentially active SNPs at IAFs ≥ 0.3 and 0.4 but not at ≥ 0.5 (although the point at 0.5 is also below 1, there are only 2 differentially active Denisovan SNPs with an IAF ≥ 0.5, explaining the large confidence intervals; S16 Fig). Of the 56 Neanderthal SNPs segregating at an IAF ≥ 0.55 considered for differential activity testing, none are differentially active, while 1 Denisovan SNP with IAF ≥ 0.55 is differentially active (of 36 SNPs considered for differential activity testing) — rs143481175, which is discussed further below and has the second strongest $\log_2$ FC in our data, -1.73, and an IAF of 0.64 in the Jacobs et al. [36] dataset.

Despite the association between IAF decile and differential activity we find no statistically significant correlation between absolute $\log_2$ FC values of differentially active sequences and IAF in either ancestry (Spearman's $\rho$ for

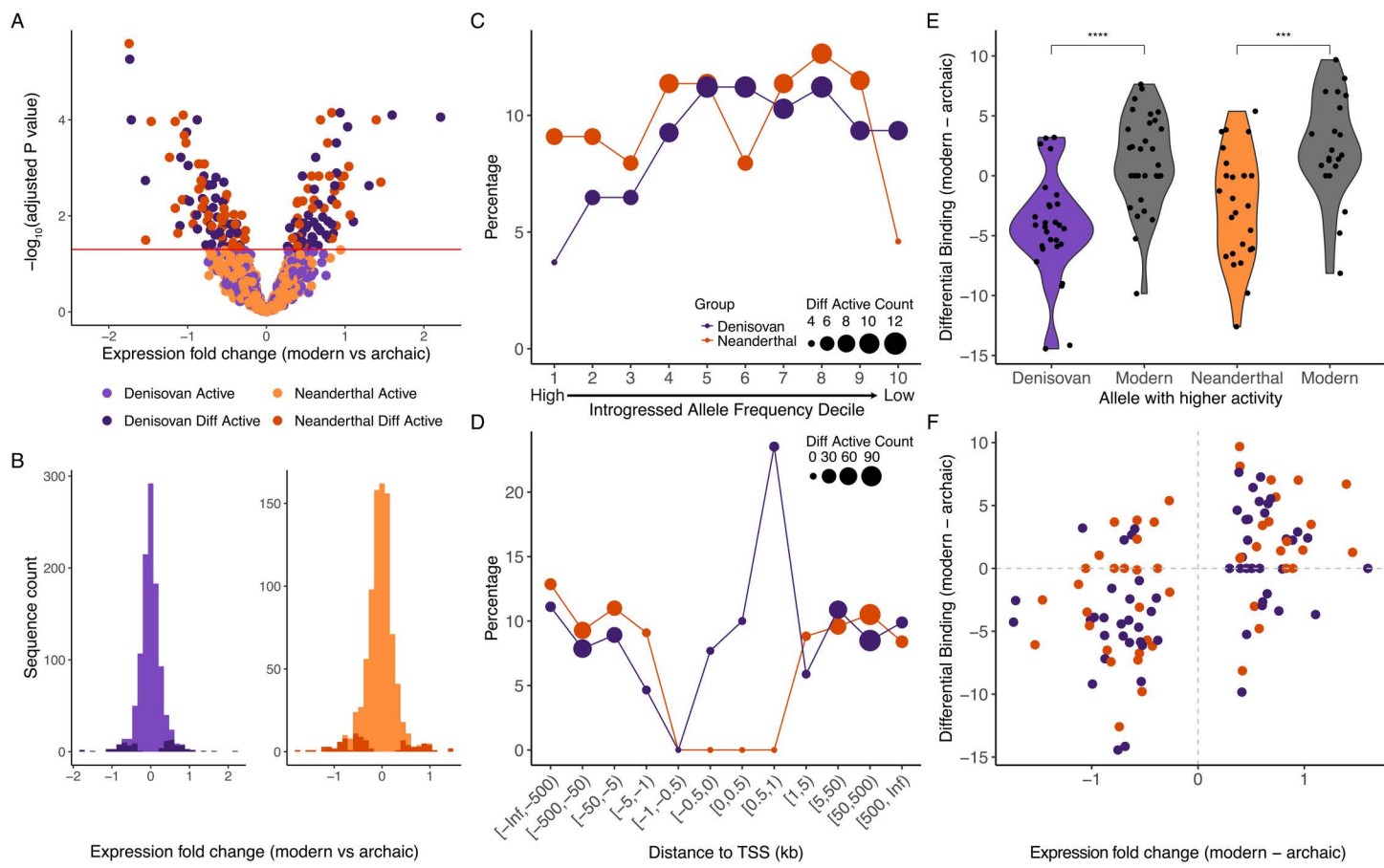

**Fig 2. Identifying differentially active introgressed SNPs. A.**) Volcano plot of results from mpralm [43]. Red line represents an adjusted p-value = 0.05. **B.**) Distribution of expression fold change in activity driven by modern and archaic alleles for Denisovan and Neanderthal test sequences. **C.**) Percentage of differentially active SNPs across IAF deciles. **D.**) Percentage of differentially active SNPs in binned distances to the transcription start site (TSS). **E.**) Distribution of differential binding scores calculated with FIMO for differentially active SNPs, polarised by effect direction. Asterisks indicate Wilcoxon test p-values: *** = p ≤ 0.001,**** = p ≤ 0.0001. **F.**) Relationship between direction of differential activity and direction of differential binding score.

Denisovan alleles = -0.192, p = 0.062; Spearman's ρ for Neanderthal alleles = 0.056, p = 0.61). We also considered how distance to TSS of genes may influence differential activity (Fig 2D). No Neanderthal allele within 1kb of an annotated TSS is differentially active, which may reflect purifying selection. However, we did observe an enrichment of differentially active Denisovan SNPs within 0.5 to 1kb downstream of the TSS compared to the overall percentage of differentially active Denisovan SNPs (binomial p = 0.008, S15 FigB), suggesting that introgression within a functional effect can be tolerated within regulatory elements.

Introgressed SNPs may alter enhancer activity through changes to transcription factor binding sites (TFBS; [10,35,44]). Indeed, we observe the highest percentage of differentially active SNPs for SNPs predicted to disrupt TF binding (S11 FigC and D). We used FIMO [45] to identify differential TF binding between the introgressed and non-introgressed alleles of a SNP. For differentially active SNPs, the direction of differential TF binding was consistent with the direction of differential activity; differentially active SNPs where the introgressed allele was more active had stronger TF binding to the introgressed allele than to the non-introgressed allele (Fig 2E; Denisovan Wilcoxon W = 730, p < 0.001; Neanderthal Wilcoxon W = 428, p < 0.001). In addition, differential binding score was correlated with estimated SNP effect size (mpralm logFC)

for both Denisovan and Neanderthal differentially active SNPs ([Fig 2F], Denisovan: r = -0.499, p < 0.001; Neanderthal: r = -0.498, p < 0.001).

Taken together, these observations again suggest that generally, selective constraint acts to keep introgressed alleles with the ability to drive differential gene regulatory activity at low allele frequencies and away from genic TSS. Furthermore, the different patterns we observe across the two sources of introgression suggests different selection pressures acting on them.

## Assessing the interactive effects of introgressed alleles

Having quantified activity across single-variant sequences, we then turned to data derived from the 1,205 haplotype-like regions. The majority (87.5%) of these contained two variant sites, meaning four distinct sequences were needed to examine all possible allelic combinations ([Fig 3A]); smaller numbers of regions contained three, four and up to six variable sites ([S4 File]), requiring $2^n$ sequences to capture all possible allelic combinations. Although all allele combinations were included in the originally synthesised oligonucleotide pool, only 6,537 haplotype-like alleles (72.99% of haplotype-like alleles in the initial library) were considered for activity and differential activity testing.

Using the same approach as above, we classified 360 Denisovan haplotype-like sequences and 192 Neanderthal haplotype-like sequences as active, where "Denisovan" and "Neanderthal" indicate introgression source—not allelic state. These sequences represented 205 distinct genomic regions (126 Denisovan, 79 Neanderthal), and ranged from carrying exclusively introgressed alleles to exclusively non-introgressed alleles. There was no significant association between the fraction of introgressed alleles in a sequence and whether it was detected as active (Denisovan p = 0.224, Neanderthal p = 0.86).

Because all variants were also tested in single-variant sequences, we wanted to ask whether we could identify more complex interactions between variants when considering them jointly. We were able to identify haplotypes for which all alleles and corresponding single-variant sequences were active, but showed variation in activity levels ([Fig 3B]). We also identified single-variant alleles that alone appear to explain haplotype activity ([Fig 3C]-[E]). For example, the introgressed Denisovan allele at chr1 89,121,892 (rs149938333) is active in a single-variant sequence, and of the corresponding haplotype-like sequences spanning chr1 89,121,792–89,121,961, only those containing the Denisovan allele at rs149938333 are active. Finally, we observed instances where reconciling the observations at the individual allele level with the haplotype level was more challenging. For example, we tested a 3-SNP introgressed Neanderthal haplotype spanning chr1 38,030,248–38,030,417 ([Fig 3E]). When tested individually, at least one allele in all three SNPs was identified as active (the human allele at rs370339360 did not meet our quality control thresholds and data from it is not reported in the figure). However, at the haplotype level, only haplotypes containing the human allele at the second SNP, rs373127224, were active, regardless of allelic state at the two other SNPs. Our study design can explain this seeming contradiction: rs369802928 and rs373127224 are located 32 bp from each other, while rs373127224 and rs370339360 are 58 bp apart. As detailed in the methods, in our library single-variant oligos are centred on the variant of interest and flanked by 75 bp of reference human genome sequence in either direction—which in this case always contains the human allele at rs373127224. Furthermore, rs373127224 is predicted to disrupt multiple transcription factor binding sites, while rs369802928 and rs370339360 do not overlap any. Thus, our data strongly suggest that out of the three SNPs tested, rs373127224 is the only one with the potential to drive activity, and highlight the value of combinatorial approaches such as these to fine map results.

We asked how often single-variant sequences showed concordant activity (active or inactive) with their corresponding haplotype-like sequences. We considered only haplotype-like sequences containing two SNPs for which all corresponding single-variant sequences were present in the final library. Additionally, we excluded haplotype-like sequences containing two archaic alleles as no corresponding single-variant sequence exists (single-variant sequences were constructed on a modern human background). This resulted in 2,115 single-variant and haplotype-like sequence pairs for testing. Across

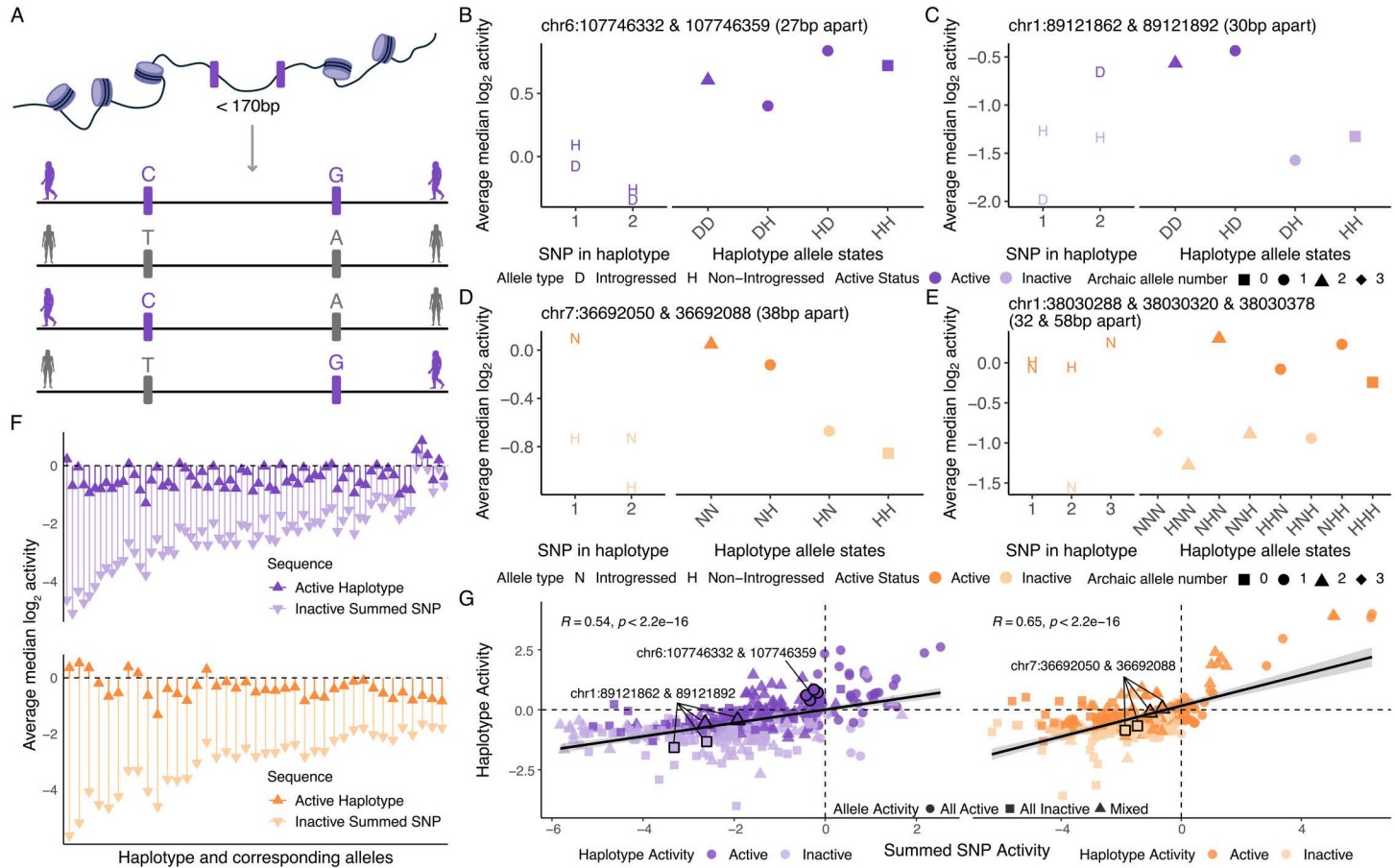

**Fig 3. Testing of haplotype-like sequences. A.**) Schematic of haplotype-like oligo design. Human silhouettes are from [phylopic.org](phylopic.org) [42]; silhouette images are by T. Michael Keesey (*Homo neanderthalensis*) and Carlo De Rito (*Homo sapiens*). B-E.) Interactive effects between single variant sequences in haplotype-like sequences. Coordinates (in hg38) show the position of the SNPs included in haplotype-like sequences. **F.**) Activity of active haplotype-like sequences for which all corresponding single SNP sequences are inactive for Denisovan (top) and Neanderthal (bottom) sequences. Plots are ordered by decreasing difference between haplotype and summed SNP activity. **G.**) Correlation between activity of haplotype-like sequences and the summed activity of the corresponding single SNP sequences for Denisovan (left) and Neanderthal (right) sequences, for haplotype-like sequences for which all corresponding single SNP sequences were present in the final library. Mixed activity of summed single SNP sequences represents cases where the single SNP sequences for a given haplotype sequence were different in their activity status. The haplotype-like sequences shown in panels B-D are outlined in black and labelled with the position of the SNPs included in the haplotype-like sequence. The haplotype-like sequences shown in panel E are not shown due to a missing corresponding single SNP sequence.

allelic states (archaic-human, human-archaic and human-human), most single-variant and haplotype-like sequence pairs shared the same active state (82–88% concordance), indicating that activity is generally robust to positional shifts of an allele within the tested sequence. However, the frequency of concordance differed significantly across allelic states ($\chi^2$ = 14.442, p = 0.0007). Sequences containing both archaic and modern alleles (archaic-human and human-archaic sequences) showed similar concordance rates (88.36% and 88.22% respectively), whereas sequences containing two modern alleles showed a decrease in concordance (82.26%). These results suggest that while single-variant and haplotype-like sequences are broadly consistent in their activity, modern human alleles may be more sensitive to positional or sequence context than archaic alleles.

For haplotype-like sequences containing two SNPs, we asked how often sequences containing two archaic alleles (archaic-archaic sequences) showed concordance in activity with sequences containing one archaic allele (archaic-human

and human-archaic sequences) to test the effect of background sequence context. Concordance in activity was high in both comparisons (archaic-archaic:archaic-human = 95%, archaic-archaic:human-archaic = 93.9%). Concordance rates did not differ significantly between the two comparisons ($\chi^2$ = 0.90, p = 0.34), suggesting that activity is largely robust to the haplotype sequence context, and the presence of an additional archaic allele does not substantially alter regulatory activity.

68 Denisovan haplotype-like sequences and 39 Neanderthal ones contained alleles that on their own did not show activity (Fig 3F) while the rest overlapped, at least partially, with some individual alleles identified as active above. Concordance between haplotype activity and the sum of their individual allelic activity was moderate (Denisovan Pearson's r = 0.53, p < 2.2 ×10$^{-16}$; Neanderthal Pearson's r = 0.64, p < 2.2 ×10$^{-16}$; Fig 3G). mpralm does not allow more than two conditions in differential activity testing. Thus, to test for differential activity of haplotype-like sequences, we performed an ANOVA on each haplotype locus, testing for differences in the activity of each allelic combination (S5 File). In this way we identified 22 haplotype sequences that showed differential activity at an FDR cutoff of 0.05 (S17 Fig). This consisted of 17 and 5 differentially active Denisovan and Neanderthal haplotypes, of which 13 and 2 respectively contained SNPs that on their own did not show differential activity.

## Predicting the effects of active and differentially active alleles

To identify potential phenotype effects of differentially active hits, we first used rGREAT [46] to investigate annotations of the genes they are predicted to regulate. As expected given our variant selection strategy, genes near Denisovan active sequences were significantly associated with Gene Ontology (GO) biological processes related to immune cell differentiation and activation, while Neanderthal active sequences were associated with terms related to cell signalling and immune cell migration (S6 File). Denisovan ancestry active alleles were additionally associated with Human Phenotype Ontology terms related to facial, skin and skeletal phenotypes, while Neanderthal active sequences were associated with terms related to dental and skeletal morphology. While we did not observe any significant associations for genes predicted to be impacted by differentially active Denisovan sequences, those near differentially active Neanderthal sequences were associated with GO biological processes such as cell signalling, immune cell migration and lipid transport. When considering overlap with GTEx v8 [47] eQTLs, 16% of differentially active variants, 89.65% of which were Neanderthal, overlapped an eQTL in any GTEx tissue. This skew is unsurprising given the under-representation of non-European individuals in GTEx but serves as further validation of the power of our assay. Similarly, in 62% of cases we observed the same direction of effect across GTEx and our data. As in Tewhey et al. [40] and Jagoda et al. [32], concordance increased to 72.88% when the weakest one-third of eQTL results were removed.

We also examined individual sequences with strong evidence of differential activity between the introgressed and non-introgressed alleles, focusing on significant signals with large predicted fold changes. Genes near these hits include *IFIH1*, *TNFAIP3*, *UBS1* and *SNX9* amongst others. The most significant result was a Neanderthal variant, rs12464349 C>T, located in intron 11 of the *IFIH1* gene. The activity of the introgressed Neanderthal T allele was markedly higher than that of the non-introgressed allele (log$_2$ FC = -1.743, p = 2.6 ×10$^{-6}$), and highly reproducible across replicates (Fig 4A, S3 File). This site is located within an introgressed haplotype defined by Jacobs et al. [36], which contains 46 testable variants of likely Neanderthal origin; of these 10 had at least one active allele in our data, but no others showed significant differential activity between the introgressed and non-introgressed allele (Fig 4B). *IFIH1* exhibits signals of adaptive introgression in Peruvians [48,49] and individuals from the Bismarck Archipelago [50]. The frequency of the T allele is 0.47 in our data [36] and 0.035 in gnomAD v 4.1; allele frequency ranges from 0.47 to 0.93 in Native American populations from HGDP-CEPH, is between 0.27 and 0.10 in 10 different Chinese ethnic minority groups and does not rise above 0.10 anywhere else in the world (S18 Fig). While this pattern might suggest misattributed Denisovan ancestry, comparison against the three existing high-coverage Neanderthal genomes confirms this group as the likely source of the sequence (S19 Fig).

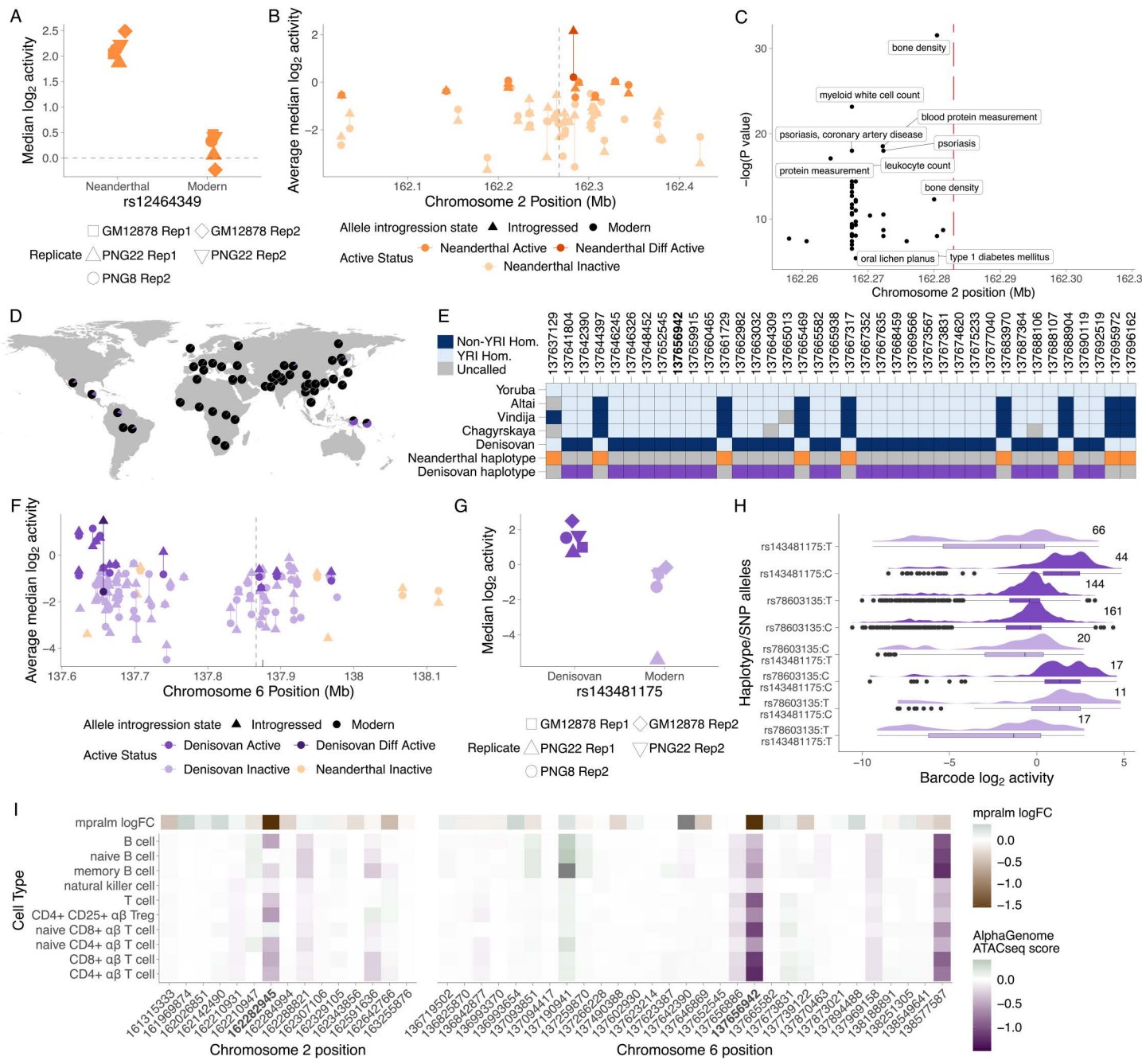

**Fig 4. Differentially active SNPs are predicted to regulate immune genes. A.)** Activity of rs12464349 alleles across replicates. **B.)** Allele activity difference for SNPs surrounding rs12464349. The transcription start site of *IFIH1* is indicated with a dashed line. **C.)** GWAS catalogue hits surrounding rs12464349 (indicated with a red line). **D.)** Introgressed allele frequency for rs143481175:C in the HGDP-CEPH populations, taken from gnomAD v 4.1. The pie centred over Port Moresby, Papua New Guinea, illustrates allele frequency in genetically Papuan individuals from Vespasiani et al. [10]. The map was created with RGGV [51] using maps from the Natural Earth Data Project [52]. **E.)** Allelic states in archaic humans and the African Yoruba (YRI) population, together with the haplotype structure across archaic marker SNPs associated with archaic haplotypes surrounding rs143481175 (bolded position) in samples from Yermakovich et al. [22]. **F.)** Allele activity difference for SNPs surrounding rs143481175. The transcription start site of *TNFAIP3* is indicated with a dashed line. The positions of rs376205580 and rs141807543 are indicated with black lines. **G.)** Activity of rs143481175 alleles across replicates. **H.)** Distribution of barcode level activity for haplotype-like oligos containing rs143481175 and corresponding single-variant oligos across replicates. Numbers indicate the number of barcodes associated with each sequence. **I.)** AlphaGenome ATAC-seq predictions for rs12464349 (left; bolded position) and rs143481175 (right; bolded position) and the surrounding SNPs tested for differential activity across immune cell types. AlphaGenome ATAC-seq score refers to the effect of the introgressed allele relative to the non-introgressed allele.

A second haplotype, labelled "Haplotype 2" in S19 Fig, segregates in East Asian groups from the 1000 Genomes but not in the Papuan samples from Jacobs et al. [36].

IFIH1 encodes MDA5, a cytoplasmic receptor that activates type I interferon signalling in response to dsRNA viral products. Gain of function mutations in IFIH1 have been robustly linked to multiple interferonopathies [53], while loss of function variants have been repeatedly associated with increased risk of SLE [54] and inflammatory bowel disease [55,56], but reduced risk of developing other autoimmune disorders, especially type 1 diabetes [57,58], suggesting a trade-off between sensitivity to antigens and autoimmunity [59], akin to that described below for TNFAIP3. Although rs12464349 is not in the GWAS catalogue, likely due to its low global allele frequency, the surrounding region harbours multiple associations with immune phenotypes, including psoriasis (Fig 4C). rs12464349 is also associated with psoriasis phenotypes in the Taiwan Precision Medicine Initiative (TPMI) [60] catalogue and is associated with terms relating to kidney function in TMPI and Biobank Japan [61].

We also identified two differentially active Denisovan SNPs predicted to regulate IRF4 (rs932780686 C>G and rs764027350 C>G). Both SNPs are located on chromosome 6 downstream of the IRF4 gene, and the Denisovan introgressed allele for both SNPs drive higher reporter gene expression than the modern alleles (rs932780686 $\log_2$ FC = -0.81, p = 4.37 $\times 10^{-3}$; rs764027350 logFC = -0.44, p = 0.0150; S20 FigA-C). IRF4 has been identified as a target of Denisovan adaptive introgression in Oceanian populations [7], and is critical for the development of lymphoid and myeloid cells, and regulates toll-like receptor signalling [62,63], another known target of Denisovan introgression [12]. In line with the role of IRF4 in lymphocyte development, GWAS catalogue hits surrounding rs932780686 and rs764027350 were linked to immune cell phenotypes, including chronic lymphocytic leukemia and lymphocyte count, as well as autoimmune diseases, such as coeliac disease, rheumatoid arthritis and type 1 diabetes. Additionally, we see GWAS terms related to dermatological phenotypes, such as alopecia, freckles, hair colour and melanoma (S20 FigD), consistent with the known association of Neanderthal introgression with skin, hair and eye pigmentation (reviewed by [64]).

## A Denisovan variant predicted to impact TNFAIP3 expression levels

Our most significant Denisovan differentially active sequence ($\log_2$ FC = -1.735, FDR-adjusted p = 5.46 $\times 10^{-6}$), and the second strongest signal in the data overall, was rs143481175 T>C, which has an IAF in Papuans = 0.64 [36] and a global AF in gnomAD v4.1 = 0.005991 (Fig 4D); this is the highest IAF amongst all differentially active sequences in our data. rs143481175 is located 162kb downstream of OLIG3 and 210kb upstream of the TNFAIP3 TSS. According to Roadmap Epigenomics data, rs143481175 falls within a predicted enhancer in the thymus, an immune tissue (ChromHMM state 7:Ehn; [37]). In the SCREEN database [65], rs143481175 is predicted to lie within a proximal enhancer that is linked to TNFAIP3 and surrounding lncRNAs, but not OLIG3, in CD4 + T-cells. Furthermore, TNFAIP3 has measurable expression in all GTEx v8 tissues, while OLIG3 has no to very low expression [47]. For these reasons we focus our analyses on the possible impact of rs143481175 on TNFAIP3 expression levels.

rs143481175 is located on an introgressed haplotype that spans roughly 120 kb on chromosome 6 (Fig 4E); the same Denisovan haplotype is present at a slightly lower mean IAF of 0.60, in the ≈ 300 samples from Yermakovich et al. [22], which are a superset of the data in Jacobs et al. [36] (S21 Fig). A second haplotype of likely Neanderthal ancestry also segregates in the same region at IAF of 0.12, while an additional Denisovan haplotype spans the TNFAIP3 TSS and is also shown in S21 Fig. The high frequency of the Denisovan haplotype, together with the presence of a second Neanderthal haplotype in the same regions, provides a clear example of a genomic region significantly shaped by archaic introgression in Papuans. Across this entire region we tested 145 single-variant sequences for activity; 134 of these were Denisovan and the remaining 11 Neanderthal. 14 of these contained at least one active allele, all Denisovan, but only rs143481175 was found to be differentially active (Fig 4F). The Denisovan introgressed rs143481175:C allele reproducibly drove higher reporter gene expression than the modern allele (T), which was not classified as active (Fig 4G).

rs143481175 was also part of a haplotype-like sequences alongside rs78603135 C>T, which is located 56 bp away from rs143481175 and has an IAF = 0.64 in our data. When tested on its own, both alleles of rs78603135 were active and we found no evidence of differential activity (log$_2$ FC = 0.003, p = 0.988). Amongst the haplotype-like sequences, only the one containing rs78603135:C and rs143481175:C was labelled as active (Fig 4H). Despite having similar median activity to the single SNP oligos, the sequence containing rs78603135:T and rs143481175:C was not labelled as active, possibly due to having low barcode counts and a clearly bimodal signal distribution (Fig 4H).

*TNFAIP3* encodes the A20 protein, an ubiquitin modifying enzyme that negatively regulates TNF-NF-κB signalling by regulating the ubiquitination of substrates in the NF-κB signalling pathway [66,67]. Polymorphisms and mutations in *TNFAIP3* have been linked to autoimmune and inflammatory disease including SLE, rheumatoid arthritis and coeliac disease (reviewed in [68]), and although rs143481175 is not in the GWAS catalogue, SNPs in the surrounding region have been associated with various autoimmune diseases including systemic lupus erythematosus (SLE), rheumatoid arthritis, coeliac disease and Sjogren syndrome (S22 Fig). The gene has been previously identified as harbouring strong signals of adaptive introgression in Papuans and Oceanians [7,14,36], and two missense mutations carried on a Denisovan intro-gressed haplotype (T108A, also known as rs376205580 A>G and I207L, also known as rs141807543 A>C) have been shown to have a slight, but significant effect on A20 activity [23]. Neither the Denisovan or modern human allele of these missense mutations were defined as active in our data, but this is to be expected, given that the experiment was designed to test for the potential to regulate gene expression.

Finally, we generated computational predictions of variant effect using AlphaGenome [69]. Because AlphaGenome struggles to predict effects on gene expression levels as distance to the target gene increases, we instead predicted ATAC-seq read coverage, as a proxy for chromatin accessibility, for all SNPs tested for differential activity across a range of immune cell types. Predicted ATAC-seq score was correlated with mpralm log$_2$ FC across immune cell types for both Denisovan and Neanderthal SNPs (Denisovan Spearman's $\rho$ = 0.112 to 0.2, p < 0.001; Neanderthal Spearman's $\rho$ = 0.114 to 0.204, p < 0.001; S23 Fig). When focusing on the regions surrounding our top two differentially active SNPs, both rs12464349 and rs143481175 had predicted ATAC-seq scores of greater magnitude than SNPs in the surrounding ± 1Mb regions (rs12464349 t = -3.7365, p = 0.002; rs143481175 t = -6.263, p = 6.95 $\times$ 10$^{-5}$, Fig 4I). The direction of allelic effect was consistent with our MPRA results, with the introgressed allele for both SNPs predicted to have greater ATAC-seq coverage than the modern human allele, providing orthogonal evidence for a role for these sites in gene regulation driven by archaic hominin introgression.

## Discussion

Introgression from Neanderthals and Denisovans has shaped modern human genomes, but the exact molecular pathways and phenotypes impacted by it remain frequently unclear—especially when considering Denisovan introgression. Intro-gression is present in regulatory regions, particularly in those associated with immune cell types [10,12–14,21,22], sug-gesting that some introgressed variants may contribute to immune phenotypes through gene regulatory changes. At the same time, multiple lines of evidence point towards selection against Neanderthal (and likely Denisovan, but this has not been as thoroughly examined) introgression, likely driven by the high genetic load and low heterozygosity in the source populations [8,9,70]. Reconciling these two observations remains challenging.

Here, we have tested the regulatory potential of over 13,000 Denisovan and 11,000 Neanderthal variants segregating at moderate or high frequencies in individuals living in Papua New Guinea today, where Denisovan introgression levels are some of the highest globally. Only a small fraction (less than 10%) of tested alleles, regardless of ancestry, were capable of driving gene expression according to our assay. This number is lower than those reported in other MPRAs that have focused on human evolution [32–34], which may reflect the biology of the types of variants being tested, but may also reflect differences in MPRA variant selection, experimental design, and analysis [71]. Our estimate is also likely to be lower than the true value, as we only tested variants in a single cell type (see, for example [34,40]), and in the absence of

any external stimulus. Nonetheless, we were able to identify introgressed and non-introgressed alleles capable of driving reporter gene expression, as well as SNPs where the two alleles drove significantly different reporter gene expression, which are strongly suggestive of functional potential *in vivo*. We also observed that the predicted direction of differential transcription factor binding was broadly consistent with the direction of observed differential activity in the MPRA, and the magnitude of both of these differences was well correlated, again supporting a functional role for these introgressed variants.

Finally, we observed moderate concordance between haplotype-like sequences and their constituent individual single-variant sequences under an assumed additive model. Our activity thresholding approach made interpreting the combinatorial effect of some variants challenging due to the fact that most MPRAs only detect activity above a baseline. Thus, when considering single-variants sequences that do not reach this cut-off, similar cDNA counts from two sequences can result in drastically different $\log_2$ activity estimates if pDNA levels are very different, even though they both indicate the same biological lack of regulatory potential. This makes additivity hard to interpret when considering inactive or lowly active sequences. Nonetheless, our results are broadly in line with those reported by other studies that have directly examined the effect of local sequence composition in expression levels using MPRAs, and found that the majority of effects between neighbouring variants are additive [72,73].

More broadly, we sought to understand the genomic forces that could impact the likelihood of an introgressed variant having functional consequences. Given conflicting reports of selection for and against introgression in regulatory regions, we examined whether distance to the nearest annotated transcription start sites was informative in this context. The percentage of active alleles did increase as distance to an annotated TSS decreased, but we did not observe an effect of allelic state (introgressed/modern) on the percentage of active alleles within 1kb of the TSS. When considering differentially active alleles, we observed no differentially active Neanderthal SNPs within 1kb of annotated TSS, although we did observe instances of Denisovan differentially active SNPs in the same range, and similar patterns between the two introgression sources at greater distances. On the whole, these trends suggest that functional, and differentially functional, introgression can be tolerated within functional regulatory elements but is nonetheless subject to some degree of constraint.

Another example of forces that could help predict function is allele frequency: an introgressed variant can rise to high frequency both in the absence of positive selection—for example, through drift or hitch-hiking—or because of it. Because introgression is frequently detected at the level of haplotypes (even SNP-by-SNP methods group variants into haplotypes to control for incomplete lineage sorting), introgression catalogues are biased towards longer introgressed segments, which makes interpreting the results of selection scans on introgressed regions challenging. In our data, we found no significant association between allele frequency and the percentage of active sequences, suggesting that allele frequency alone is not enough to define or predict functional introgression. However, we did observe some evidence of constraint against differential activity at high IAF, particularly for Denisovan variants, and to some extent on Neanderthal variants, with no Neanderthal allele segregating at an IAF greater than 0.55 deemed differentially active. Yet despite this, the two strongest signals in our data came from alleles with high IAF: rs12464349, within an intron of *IFIH1* has an IAF of 0.47, and rs143481175, near *TNFAIP3*, has an IAF of 0.64 in the Jacobs et al. [36] dataset, and one of the highest IAFs in the entire dataset. Notably, other variants in both of these genes have been implicated in evolutionary trade-offs between increased auto-immunity and decreased immune surveillance [23,59].

In our data, the Denisovan C allele at rs143481175 drove significantly higher levels of reporter gene expression than the modern human allele; suggesting it may increase expression levels of *TNFAIP3 in vivo* and possibly of A20 (the protein encoded by *TNFAIP3*). This increase may lead to increased downregulation of NF-κB signalling pathways following immune activation, which could be beneficial in reducing chronic inflammation and autoimmunity—at the cost of decreased immune system responsiveness. Notably, the direction of this effect is opposite to that of two non-synonymous mutations introgressed from Denisovans (T108A and I207L) in *TNFAIP3*. Both of these mutations are common in Oceanic

populations, including those that show otherwise low levels of introgression, but almost absent west of Wallace's line [23], suggesting positive selection drove their spread beyond genetically Papuan individuals. These variants give rise to a A20 protein with partial phosphorylation defects that lightly reduce the inhibitory activity of the protein, increasing immunity and resistance to coxsackievirus without causing spontaneous inflammatory disease that is associated with more severe loss of function [23]. As such, the function of the A20 protein may be balanced by the effects of rs143481175 and the T108A/I207L alleles.

Unfortunately, the staggeringly poor representation of individuals carrying Denisovan introgression—and more broadly, of non-European genetic ancestries—in existing public genomic resources means further experimental validation of these results in orthogonal datasets is extremely challenging. For example, when looking for overlap between variants in the GWAS catalogue [74] and our active SNPs, only 2 Denisovan and 5 Neanderthal SNPs were present in the GWAS catalogue. We were able to identify 274 active SNPs that are also eQTLs in GTEx v8 [47], the majority of which were Neanderthal hits, as expected due to the disproportionate number of European donors in GTEx. These disparities have been well-documented before (e.g., [75,76]); both logistic and ethical challenges in generating these datasets outside the Global North [77] mean that they will be hard to redress in the short and middle term, at least at comparable scales to existing resources. Within this set of overlapping SNPs, however, the direction of effect was generally consistent between GTEx and our MPRA, confirming the value of MPRAs for large-scale variant effect testing and fine-mapping. Thus, MPRAs and other modalities of multiplex assays of variant effect (MAVEs), alongside computational predictors of variant effects, emerge as leading candidates to drive our understanding of the functional consequences not only of introgressed variants, but of genetic variation found in groups and individuals who are poorly represented in existing resources.

## Materials and methods

### Ethics approvals and human samples

The two Papuan cell lines, PNG8 and PNG22, were established from healthy adult donors sampled at the University of Papua New Guinea, in Port Moresby, Papua New Guinea. Collection of blood was coordinated by Dr Christopher Kinipi (Director of Health Services at the University of Papua New Guinea) and approved by the Medical Research Advisory Committee of the National Department of Health of the Government of Papua New Guinea (permit number MRAC 16.21), by the University of Melbourne's Human Research Ethics Committee (approvals 1851585.1 and 26981), and by the French Ethics Committees (Committees of Protection of Persons 25/21_3, n°SI:21.01.21.42754). Permission to conduct research in Papua New Guinea was granted by the National Research Institute of Papua New Guinea (permit 99902292358), with full support from the School of Humanities and Social Sciences, University of Papua New Guinea. These approvals commit our team to following all ethical guidelines mandated by the government of Papua New Guinea. All individuals gave their full informed written consent to participate in the study.

### Variant selection and library design

To select variants for testing in the MPRA, we selected archaic variants segregating in genetically Papuan individuals from Jacobs et al. [36] (for detailed information on defining archaic variants, see Vespasiani et al. [10]), if they were located in DNase I hypersensitivity sites or Tn5 transposase accessible regions in different immune cell types from the Roadmap Epigenomics Consortium [37], the ENCODE project [38] and [39]. We also included a set of archaic SNPs predicted to disrupt transcription factor binding site [10]. SNPs annotated in at least one active chromatin state within monocytes, LCLs, B or T cells in Epigenome Roadmap data [37] were retained for testing (S1, S2 Figs). SNPs may be assigned to multiple annotations within and/or across cell types. To prioritise variants of evolutionary and phenotypic importance, we additionally filtered for variants with a IAF of $\geq$ 0.15. Introgressed allele frequency was calculated as the number of observations of the introgressed allele divided by the number of observations for all alleles.

We additionally included two sets of control SNPs previously found to be active in MPRAs in LCLs [40] and in HepG2 cells [41]. For SNPs from Tewhey et al. [40], we downloaded the full set of variants tested (https://www.cell.com/cms/10.1016/j.cell.2016.04.027/attachment/19be3b82-539c-4548-b522-cac927d98d55/mmc3.xlsx) and selected those with an rsid and absolute $\log_2$ allelic skew combined across the tested LCLs greater than 0.81. Where multiple alternative alleles were tested for the same rsid, we selected the allele with the maximum absolute $\log_2$ allelic skew. For SNPs from Arensbergen et al. [41], we randomly selected 200 SNPs that were active in HepG2 cells, but not K562 cells.

Finally, we also designed a set of 300 random scrambled sequences to act as negative controls. For these random sequences, we confirmed the absence of any DNA motifs in HOCOMOCO v11 [78] and JASPAR 2020 [79] with FIMO [45] and the absence of matches to known human genomic DNA sequences with BLAT [80]. We designed 170 bp oligos centred on the selected SNPs, flanked by 15 bp of an adapter sequence. Where two or more SNPs where located within 170 bp of each other, we designed haplotype-like oligos containing all possible allelic combinations of the SNPs. We removed oligos containing SfiI and BsiWI restriction sites as these would be lost downstream in the cloning process. In total, our MPRA library included 61,812 oligos, testing a total of 25,869 introgressed SNPs (S1 File).

## Oligosynthesis and MPRA vector assembly

Oligos were synthesised by Twist Biosciences as 200 bp single-stranded DNA sequences containing 170 bp of genomic context centred on the variant of interest and 15 bp of adapter sequence at either end (5'ACTGGCCGCTTGACG [170 bp oligo] CACTGCGGCTCCTGC3'). We converted oligos to double-stranded DNA in a low cycle PCR (KAPA HiFi HotStart PCR Kit) with primers designed to bind to the adapter sequences (F_adapter and R_adapter, S7 File). We next added 15 bp random barcodes, SfiI and BsiWI restriction sites, HiFi cloning sites and a buffer sequence to the oligos in another PCR reaction (NEB Q5 High-Fidelity 2x Master Mix; F_PCR and Barcoding_R, S7 File). Barcoded oligos and pMPRA1 plasmid were digested with SfiI (NEB) and the digested oligos ligated into the pMPRA1 plasmid with T4 ligase (NEB). pMPRA1 was a gift from Tarjei Mikkelsen (Addgene plasmid # 49349, [81]). The resulting plasmid library was transformed into 10-beta electrocompetent E. coli (NEB) using a BioRad GenePulsar (2.0 kV, 200 Ω and 25 µF) and allowed to grow overnight in five flasks containing 500ml LB-SeaPrep agarose (Lonza) culture [82] and ampicillin at a final concentration of 0.1mg/ml. Colonies were pelleted via centrifugation and giga-prepped (Zymo). To associate each barcode with its corresponding oligo, we amplified a fragment containing the oligo and barcode from each plasmid with PCR (NEBNext Ultra II Q5 Master Mix). Fragments were sequenced by Azenta with 150 bp paired end reads on Illumina Novaseq. We used the MPRAmatch pipeline (https://github.com/tewhey-lab/MPRA_oligo_barcode_pipeline, adapted with custom scripts) to associate barcodes with oligos (S8 File).

The plasmid library was digested with BsiWI (NEB) and GFP (IDT) was cloned into the plasmid vector with HiFi assembly (NEBuilder HiFi DNA Assembly MasterMix). The final plasmid library was transformed into 10-beta electrocompetent E. coli (NEB) using a BioRad GenePulsar (2.0 kV, 200 Ω and 25 µF) and allowed to grow overnight in four flasks containing 500ml LB with ampicillin (final concentration = 0.1mg/ml). Colonies were pelleted via centrifugation and giga-prepped (Zymo). The concentration of the resulting plasmid preparation was increased with ethanol precipitation.

## Cell culture and transfection

Three lymphoblastoid cells lines, GM12878 (Coriell), PNG22 and PNG8 were grown in suspension in RPMI (Gibco) supplemented with 10% FBS (Scientifix), 1% NEAA (Gibco) and 1% Glutamax (Gibco), maintaining a density of 0.2-0.5 × 10⁶ cells per mL until there was a total of 120 × 10⁶ cells per line, except for GM12878 replicate 1, where we grew 160 × 10⁶ cells. The day prior to transfection with a Nucleofector II (Lonza), cells were seeded at a concentration of 0.5x10⁶. The day of transfection, cells were counted using a Countess 3 (Thermo Fisher) and pelleted at 500xg for 5 minutes. Cells were resuspended in supplemented Nucleofection solution V. The plasmid library was added to this solution, and 100$\mu$l cell-DNA suspension was transferred to a nucleofection cuvette (each cuvette contained 5 x 10⁶ cells and 40µg plasmid

 

library). The cuvette was inserted into the Nucleofector II device and program X-001 applied. Immediately after, 500$\mu$l of prewarmed culture medium was added to the cuvette. The nucleofected cell suspension was then transferred to a T75 flask and incubated for 24 hours. This process was repeated until all cell-DNA suspension has been transfected (12–16 reactions per replicate). 24 hours post-transfection, cells were pelleted for 5 minutes at 500xg. The media was removed and cell pellets resuspended in 2ml DPBS. 90% of the suspension was taken for RNA extraction and 2 x 5% was taken for DNA extraction. Cells were spun down again at 500xg for 5 minutes. The DPBS was removed and the cell pellets frozen at -80 C for storage. Two replicates were performed per cell line, with $60 \times 10^6$ cells for all replicates except GM12878 replicate 1, where we transfected $80 \times 10^6$ cells.

## DNA/RNA extraction and cDNA synthesis

Plasmid DNA (pDNA) was extracted from the cell pellets using Qiagen's DNeasy Blood and Tissue Kit following manufacturer's protocols, performing an optional digestion with RNase A (Qiagen). RNA was extracted from the cell pellets using Qiagen's RNeasy Midi kit, performing an optional on-column DNase digestion (Qiagen). RNA extractions were treated with Turbo DNase (Thermo Fisher) to remove any DNA carryover. First-strand cDNA was synthesised with Superscript IV (Thermo Fisher) using a primer designed to bind to the buffer sequence (reverse_transcription_primer, S7 File). To quantify the number of cycles required for library amplification, we performed qPCR (NEBNext Ultra II Q5 master mix 2x) for both pDNA and first-strand cDNA using primers binding to GFP and the buffer sequence (GFP_F and Buffer_sequence_R, S7 File). Using the Ct values determined from qPCR, the barcode region of the cDNA and pDNA libraries was amplified with PCR (NEBNext Ultra II Q5 master mix 2x; GFP_F and Buffer_sequence_R, S7 File). PCR products were cleaned using Qiagen's MinElute kit, and checked for product size and concentration with Agilent Tapestation. Samples were initially sequenced to 20M paired end reads on Illumina NovaSeq by Azenta. After confirming the quality of the sequencing library, the samples were resequenced to 240M paired end reads. An additional replicate (PNG8 rep1) did not have enough cDNA to be sequenced, so it was omitted from all analyses.

## Data analysis

Adapter sequences were removed from reads from both rounds of sequencing with Cutadapt v4.8 [83]. Paired end 150 bp reads were merged with Flash v2.2 (flags: -r 150 -s 30 -t 25) [84] and reads from the two rounds of sequencing merged. Reads were passed to the MPRAcount pipeline (https://github.com/tewhey-lab/MPRA_oligo_barcode_pipeline, adapted with custom scripts) to count the number of each barcode in the cDNA and pDNA libraries (S24 Fig). Following methods from Uebbing et al. [33], cDNA and pDNA counts were normalised by library size and $\log_2$ transformed (S9 File). Barcodes with an average pDNA $\log_2$ CPM < –5 across replicates were removed (S25 Fig). We also removed oligos for which there were fewer than 10 associated barcodes. For each barcode, we calculate its 'activity' as the ratio of cDNA counts over pDNA counts. To define active sequences, we compare the activity distribution of all barcodes for an oligo against the mean activity per replicate using a one-sided $t$ test. P-values were corrected with Benjamini-Hochberg multiple testing correction, and we define a fragment as active if it has a corrected p-value < 0.05 in at least two replicates. Sequences were defined as European specific if their corrected p-value was < 0.05 in both European replicates and > 0.05 in all three Papuan replicates. Sequences were defined as Papuan specific if their corrected p-value was < 0.05 in at least two of the three of the Papuan replicates and > 0.05 in both European replicates.

All sequences with at least one active allele were then tested for differential activity. Where relevant, we compare active alleles to their inactive counterparts. We take two approaches to differential activity testing. First, to test for differential activity between two alleles of a SNP, we used the mpra v1.26 R package [43], which makes use of the voom framework [85] to model the properties of count data. As recommended by Myint et al. [43] cDNA and pDNA counts for each oligo were summed across barcodes and values for all five replicates passed on to mpralm. The model included terms for allelic

state (introgressed or not) and cell line genetic background (Papuan or not). We define sequences as differentially active between the introgressed and modern alleles if it has a Benjamini-Hochberg adjusted p-value <0.05.

To test for differential activity of haplotype-like sequences, we apply ANOVA to test for differences in activity of each oligo per replicate. Haplotype-like oligos were defined as differentially active if they had a Benjamini-Hochberg adjusted p-value <0.05 in at least two replicates. Activity testing and subsequent analysis was performed in R v.4.4.0.

## SNP annotation

To assess whether allele frequency influences activity, allele frequencies were calculated as in Vespasiani et al. [10]. For fair comparison of the frequency of introgressed and modern alleles, for each introgression source (Denisovan/Neanderthal) and allelic state (introgressed/modern), we divide sequences into allele frequency deciles, and calculate the percentage of active alleles within each decile. Similarly, to assess whether allele frequency influences differential activity, we divide sequences into introgressed allele frequency deciles, and calculate the percentage of differentially active sequences within in decile.

To assess how distance to TSS influences activity, we used the *getRegionGeneAssociations* command in rGREAT v2.6 [46] to link sequences to genes they putatively regulate. We ran rGREAT on our active and differentially active hits using hg38 coordinates (converted from hg19 coordinates with rtracklayer v1.64 [86]), separately for Denisovan and Neanderthal hits, with the *great()* command, specifying mode = "twoClosest" and tss source = "Gencode v40". We binned distances to the TSS, and calculated the percentage of active and differentially active sequences within each distance bin.

To identify enriched Gene Ontology and Human Phenotype Ontology terms, we used rGREAT to test for enrichment of active and differentially active hits in MSigDB GO and HPO gene sets, using the set of sequences tested for activity and differential activity as background sets. Enrichments were considered significant at FDR < 0.05.

To assess the disruption of TFBS, all sequences in the MPRA library were first trimmed to the middle 40 bp. The trimmed list of sequences was passed to FIMO [45] to identify matches with the HOCOMOCO v11 core collection [78], regardless of their expression in LCLs or immune cells, and only retaining matches with p-value <0.0001 in the output. We filtered for Denisovan and Neanderthal sequences where one allele has a p-value $<10^{-4}$ and the other has a p-value $<10^{-3}$. For these sequences, we calculate a differential binding score between the introgressed and the modern allele by taking the difference between FIMO binding scores for the two alleles. Where a SNP was predicted to disrupt multiple TFBS, we take the motif with the largest absolute differential binding score.

To search for SNPs in the GWAS catalogue, we used v1.0.2 of the GWAS catalogue (downloaded from https://www.ebi.ac.uk/gwas/docs/file-downloads). We searched for matching rsIDs between our MPRA SNPs and the GWAS catalogue. To define GWAS SNPs in the regions surrounding our differentially active hits, we searched the GWAS catalogue for SNPs within 50kb of each differentially active SNP.

To search for eQTLs overlapping and surrounding our hits, we downloaded GTEx v8 [47] significant variant-gene pairs for all tissues (https://gtexportal.org/home/downloads/adult-gtex/qtl). We merged eQTLs for all tissues, and overlapped these with the set of variants tested in our MPRA by chromosome, hg38 coordinates, the reference and alternative allele. To examine concordance between our data and GTEx, we transformed the mpralm effect size to reflect the effect of the alternative allele against the reference allele, and compare the direction of this effect to the slope of the eQTL. To assess concordance after removing weak effects, we removed the smallest one-third of absolute effect sizes, and recalculated concordance.

We generated computational predictions of variant effects for all SNPs tested for differential activity with AlphaGenome [69] using the publicly available API and the *dna_model.score_variant* command. We used 500kb surrounding the variant of interest as input. Raw AlphaGenome scores represent the difference between the alternative and reference allele, and so were transformed to reflect the difference between the introgressed and non-introgressed allele. For our top

differentially active SNPs, we tested for differences in AlphaGenome ATAC-seq score across cell types compared to the surrounding SNPs with a one sided t-test (alternative = "lower").

## Supporting information

**S1 Fig. Number of Denisovan SNPs in initial library in chromHMM chromatin states across immune cell types.**
(EPS)

**S2 Fig. Number of Neanderthal SNPs in initial library in chromHMM chromatin states across immune cell types.**
(EPS)

**S3 Fig. Cell type specificity of annotations used for SNP selection for A) Denisovan and B) Neanderthal SNPs.**
(EPS)

**S4 Fig. Distribution of IAF for A) Denisovan and B) Neanderthal measured alleles.**
(EPS)

**S5 Fig. Distribution of barcodes per oligo post filtering.** The number of barcodes is calculated as the number of unique barcodes with average pDNA $\log_2$ CPM > -5 across replicates.
(EPS)

**S6 Fig. Scatterplots of cDNA and pDNA counts per replicate.** Active sequences are highlighted in colour. Positive control 1 and 2 sequences were previously found to be active by Tewhey et al. [40] and Arensbergen et al. [41], respectively.
(EPS)

**S7 Fig. Sharing of activity across replicates.** A) Denisovan, B) Neanderthal, C) positive controls from Tewhey et al. [40], D) positive controls from Arensbergen et al. [41] and E) random scrambled sequences. Sequences were labelled as active if active in at least two replicates.
(EPS)

**S8 Fig. Spearman's correlation of A) activity calculated at the summed barcode level and B) median activity across replicates for B) all sequences, C) archaic sequences, D) positive control sequences and E) random scrambled sequences.**
(EPS)

**S9 Fig. Spearman's correlation of median activity across replicates for all sequences after filtering for sequences with at least A) 10, B) 20, C) 30, D) 40, E) 50, F) 75 and G) 100 barcodes. The number of sequences retained after each barcoding threshold is displayed.**
(EPS)

**S10 Fig. Association between activity status and GC content for Denisovan and Neanderthal sequences.**
(EPS)

**S11 Fig. Percentage A-B) active sequences and C-D) differentially active SNPs across the annotations used to define SNPs for testing.** Active and open refer to active chromatin and open chromatin, respectively.
(EPS)

**S12 Fig. Enrichment of active sequences across SNP annotations, stratified by whether that annotation is cell type specific or shared.**
(EPS)

**S13 Fig. Percentage A) active by allele frequency bins and B) differentially active hits by introgressed allele frequency bins.**
(EPS)

**S14 Fig. Percentage active by A) allele frequency decile and B) distance to the TSS.** For each decile/bin, we tested whether the percentage of active alleles is different from the overall percentage of active Denisovan or Neanderthal alleles (indicated by the dashed lines) with binom.test() and show the resulting confidence intervals.
(EPS)

**S15 Fig. Percentage differentially active SNPs by A) allele frequency decile and B) distance to the TSS.** For each decile/bin, we tested whether the percentage of differentially active SNPs is different from the overall percentage of differentially active Denisovan or Neanderthal SNPs (indicated by the dashed lines) with binom.test() and show the resulting confidence intervals.
(EPS)

**S16 Fig. Depletion of differentially active SNPs at varying introgressed allele frequency thresholds.** Odds ratios calculated with Fisher's exact test.
(EPS)

**S17 Fig. Activity of differentially active haplotypes (top row) and the activity of their corresponding single SNP sequences (bottom row) for Denisovan (left) and Neanderthal (right) sequences.** Haplotype-like sequences are coloured by the percentage of introgressed alleles within the haplotype. Pairs of shapes for single variant sequences represents the two alleles of any given SNP. The Denisovan haplotype shown in Fig 3C and the Neanderthal haplotype in Fig 3E are shown with black outlines.
(EPS)

**S18 Fig. IAF for rs12464349:T in the HGDP-CEPH populations, taken from gnomAD v 4.1.** The pie centred over Port Moresby, Papua New Guinea, reports allele frequency in genetically Papuan individuals included in Vespasiani et al. [10] instead. The map was created with RGGV [51] using maps from the Natural Earth Data Project [52].
(EPS)

**S19 Fig. Allelic states in archaic humans and the African Yoruba (YRI) population, together with the haplotype structure across archaic marker SNPs associated with archaic haplotypes surrounding rs12464349 in samples from Yermakovich et al. [22].**
(EPS)

**S20 Fig. Two differentially active SNPs predicted to regulate *IFR4*.** A). Activity of rs932780686 alleles across replicates. B). Activity of rs764027350 alleles across replicates. C.) Allele activity difference for SNPs surrounding rs932780686 and rs764027350. The transcription start site of *IRF4* is indicated with a dashed line. D.) GWAS catalogue hits surrounding rs932780686 (indicated with a dashed line) and rs764027350 (indicated with a solid line).
(EPS)

**S21 Fig. Frequency distribution of archaic haplotypes across the 1MB region surrounding rs143481175 in the Papuan dataset from Yermakovich et al. [22].** Haplotypes are classified as Neanderthal and Denisovan based on their sequence similarity with Neanderthals and the Denisovan. The position of rs143481175 is indicated by the dashed line.
(EPS)

**S22 Fig. GWAS catalogue [74] hits surrounding rs143481175.** The position of rs143481175 is indicated by the red dashed line.
(EPS)

**S23 Fig. Association between mpralm logFC and AlphaGenome ATAC-seq score across immune cell types.**
(EPS)

**S24 Fig. Distribution of barcodes per oligo after first round sequencing.** The median number of barcodes per oligo is indicated in red, the mean per oligo in blue. Barcode counts per oligo were generated with the MPRAmatch pipeline (https://github.com/tewhey-lab/MPRA_oligo_barcode_pipeline).
(EPS)

**S25 Fig. Density plots of $\log_2$ pDNA barcode counts per replicate.** We excluded all barcodes with a $\log_2$ CPM below -5, indicated by the red line.
(EPS)

**S1 File. All sequences included in MPRA library.**
(XLSX)

**S2 File. Output of activity testing of single-variant sequences.**
(XLSX)

**S3 File. Output of differential activity testing of single-variant sequences.**
(XLSX)

**S4 File. Number of alleles per haplotype.**
(XLSX)

**S5 File. ANOVA p-values for differential activity testing of haplotype sequences.**
(XLSX)

**S6 File. Significant results of rGREAT enrichment testing.**
(XLSX)

**S7 File. Primer sequences used in MPRA cloning.**
(XLSX)

**S8 File. Output of MPRAmatch pipeline for creating barcode-oligo associations.**
(XLSX)

**S9 File. Normalised and $\log_2$ transformed output of MPRAcount pipeline.**
(ZIP)

## Acknowledgments

We thank other members of the Gallego Romero lab, Pradiptajati Kusuma and Mait Metspalu for valuable comments on the manuscript, and sample donors at the University of Papua New Guinea for their contribution to this work.

## Author contributions

**Conceptualization:** Davide M Vespasiani, Irene Gallego Romero.

**Data curation:** Maddy Comerford, Danat Yermakovich.

**Formal analysis:** Maddy Comerford, Davide M Vespasiani, Michael Dannemann.

**Funding acquisition:** François-Xavier Ricaut, Irene Gallego Romero.

**Investigation:** Maddy Comerford, Davide M Vespasiani.

**Methodology:** Maddy Comerford, Davide M Vespasiani, Navya Shukla, Laura E Cook.

**Resources:** Danat Yermakovich, Michael Dannemann, Matthew Leavesley, Christopher Kinipi, François-Xavier Ricaut, Nicolas Brucato.

**Visualization:** Maddy Comerford, Michael Dannemann.

**Writing – original draft:** Maddy Comerford, Irene Gallego Romero.

**Writing – review & editing:** Maddy Comerford, Davide M Vespasiani, Navya Shukla, Laura E Cook, Danat Yermakovich, Michael Dannemann, Matthew Leavesley, Christopher Kinipi, François-Xavier Ricaut, Nicolas Brucato, Murray P Cox, Irene Gallego Romero.

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
