## [Decision Letter · Decision Letter 0]

19 Aug 2025

PGENETICS-D-25-00637

Mapping the gene regulatory landscape of archaic hominin introgression in modern Papuans

PLOS Genetics

Dear Dr. Gallego Romero,

Thank you for submitting your manuscript to PLOS Genetics. After careful consideration, we feel that it has merit but does not fully meet PLOS Genetics's publication criteria as it currently stands. Therefore, we invite you to submit a revised version of the manuscript that addresses the points raised during the review process.

Please submit your revised manuscript within 60 days Oct 18 2025 11:59PM. If you will need more time than this to complete your revisions, please reply to this message or contact the journal office at plosgenetics@plos.org. Please include the following items when submitting your revised manuscript:

We look forward to receiving your revised manuscript.

Kind regards,

Francesca Luca

Guest Editor

PLOS Genetics

Justin Fay

Section Editor

PLOS Genetics

Aimée Dudley

Editor-in-Chief

PLOS Genetics

Anne Goriely

Editor-in-Chief

PLOS Genetics

**Additional Editor Comments:**

As you will see, the reviewers were overall excited about your study and provided very detailed comments, which should be helpful and guide you in preparing a revised version of the manuscript.

**Journal Requirements:**

https://journals.plos.org/plosgenetics/s/submission-guidelines#loc-parts-of-a-submission

4) We notice that your supplementary Figures are included in the manuscript file. Please remove them and upload them with the file type 'Supporting Information'. Please ensure that each Supporting Information file has a legend listed in the manuscript after the references list.

Potential Copyright Issues:

- Figures 4D and S9. Please (a) provide a direct link to the base layer of the map (i.e., the country or region border shape) and ensure this is also included in the figure legend; and (b) provide a link to the terms of use / license information for the base layer image or shapefile. We cannot publish proprietary or copyrighted maps (e.g. Google Maps, Mapquest) and the terms of use for your map base layer must be compatible with our CC BY 4.0 license.

**Reviewers' comments:**

Reviewer's Responses to Questions

**Comments to the Authors:**

Reviewer #1: The manuscript « Mapping the gene regulatory landscape of archaic hominin introgression in modern Papuans » by Comerford et al uses plasmid-based Massively parallel reporter assays to test the effect on lymphoblastoid expression of 25,869 introgressed alleles segregating at high frequency (>15%) in present-day Papuans. The manuscript is of high quality and combines general observations on the phenotypic effect of archaic introgression, with the identification of specific variants with strong effects on immune gene expression. The manuscript reads very well and is highly relevant to researchers interested in either genetics of gene expression, or population genetics.

I particularly enjoyed the results on the IFIH1 variant which I find to be a compelling example of how adaptive introgression has impacted the human immune system worldwide.

***Major comments :***

1/ Several analyses in the manuscript rely on comparing the percentages of active sequences across distinct classes of variants (eg. based on introgressed allele frequency or distance to TSS), however several comparisons are made without reporting significance, or confidence intervals.

For instance:

• Lines 177-178, the authors state that ‘More Denisovan active sequences were specific to the Papuan replicates (91 [out of 128] ) than Neanderthal active sequences (80 [out of 145])’. I’m not sure this difference is statistically significant. A Fisher’s exact test seems to give p=0.2.

• Lines 196-197: “[the percentage of sequence located within 1kb of the nearest TSS that are active] is consistently lower for introgressed alleles than for AMH ones.” Could the authors provide a formal p-value?

• lines 213-216, the authors seem to oppose the pattern observed for Denisovan and Neanderthal alleles, yet they do not formally test for differences in the proportion of differentially active variants, with either one or both alleles between Denisova- and Neanderthal- introgressed SNPs.

• Lines 234-236, the authors report higher percentage of differentially active SNPs among Denisovan-introgressed variants <1kb downstream of the promoter, but do not report significance.

More generally, several plots are hard to interpret due to the lack of confidence intervals, including figures 1E, 1F, 2C, 2D & S8. Confidence intervals could be easily computed with the binom.test R function.

2/ On a related note, the authors’ interpretation regarding the link between introgressed allele frequency and functionality is unclear, as several seemingly contradictory statements appear throughout the manuscript.

• Lines 218-223, the authors claim that « Denisova and Neanderthal alleles display markedly different patterns », although as mentioned above, there is no statistical test to support a difference in the proportion of differentially active alleles between Denisovan and Neanderthal allele SNPs segregating at high/low frequency.

• A few lines later (lines 229-230), the authors report a lack of significant correlations between introgressed allele frequency and logFC values (despite a suggestive trend for Denisovan-introgressed alleles at p=0.06 which they do not explicitly comment).

• In the discussion, (lines 442-445) they mention evidence of constraint against differential activity at high IAF for Denisova variants (despite the statistical significance of this observation has not been formally established, aside from the borderline signal at p=0.06) and discuss the lack of differentially active Neanderthal alleles at IAF>0.55, yet it is unclear whether the low number of Neanderthal alleles with IAF>0.55 could explain such a lack of active alleles.

• In the abstract, the authors report no impact of introgressed frequency on activity, which seems in contradiction with the claims in lines 442-445.

I would suggest that the authors add an additional analysis where they test, for different frequency thresholds (eg. IAF>0.3, 0.4, 0.5, 0.55 & 0.6), how the percentage of differentially active sequences varies between alleles introgressed at high or low frequency (separately for Neanderthal and Denisovan alleles). Fisher’s exact test should be enough. If there is indeed a trend toward lower functionality at higher frequency for Neandertal/Denisovan-introgressed alleles, this analysis should grant them enough power to detect it.

I would then encourage the authors to revise the text, in accordance with these new results to clarify their conclusions and avoid contradictory statements.

3/ I wonder to what extent the association that the authors find with the distance to TSS (Figure 1F) could be biased by the local GC content which tends to increase near the TSS? In my own experience, GC content can be linked with overall activity, with GC-rich regions often being more active. Have the authors looked into this? Do they see the same trend if activities are adjusted for GC content?

GC content can also be correlated with the number of barcodes associated with a specific sequence in the plasmid library, and thus the power to detect differentially active sequences. I don’t think this could explain the results from Fig 2D, but it could contribute to adding noise in the data.

4/ When discussing specific SNPs (eg. lines 325-327, or lines 379-389), the authors report a lack of associations for the reported SNPs in the GWAS catalog and instead report nearby GWAS associations as a proxy for gene function. While this approach is indeed justified by the general bias of GWAS toward Europeans, which prevents the interrogation of Papuan-specific variants, the authors could go a bit further by interrogating non-European Biobanks such as Biobank Japan (https://pheweb.jp) or the Taiwan Precision Medicine Initiative (https://pheweb.ibms.sinica.edu.tw), where some of these variants are present at reasonable frequency.

5/ While the analyses presented in Figure 3 are definitely interesting and appear rather sound, I found them difficult to understand. In particular, I regretted that the authors did not provide plots showing for specific loci the activity (y axis ) of each tested haplotype (x-axis), to highlight how the effect of the various SNPs combines in either an additive or non-additive manner.

Could the authors make plots (could be on the main figure, or in supplementary) where they focus on a single locus (i.e. one column of Figure 3D) to better illustrate the phenomenon they are discussing ?

To clarify my idea, I attached a toy example (Example_figure1.pptx) of the kind of plots I would expect. These could be shown in the supplement or in place of the current figure 3D (which could be moved to supplements to provide a fuller picture on haplotypic effects for the interested reader).

6/ It seems that the assignment of target genes is mainly performed based on distance and known gene annotations (for TNFAIP3). Did the authors consider validating the targets of the identified variants through CRISPR or equivalent? Alternatively, they could use Promoter-capture HiC data from (Javierre et al., Cell, 2016), or HiChIP data from (Chandra et al., Nat Genet., 2021). They could also query the SCREEN database (https://screen.wenglab.org/) to identify putative enhancer-gene links based on experimental evidence (Intact Hi-C Loops, ChIA-PET Interactions, CRISPRi-FlowFISH), or computational predictions (ABC, EPIraction, GraphRegLR, or rE2G).

7/ Although I would not make this a strict requirement, I think the manuscript would be strengthened if the authors could validate some of their main findings (eg. rs12464349, & rs143481175) through standard luciferase assays to demonstrate that their main findings are robust and replicate with a different technique.

*** Minor comments: ***

1/ line 54: the authors report that Denisova introgression accounts for up to 5% of genomes of Papuans and Near Oceanic Ancestry. I believe this estimate is outdated and more recent studies (eg. Choin et al, Nature, 2021) have estimated that while Papuans indeed carry ~5% of archaic DNA, only 3% of their DNA is inherited from Denisovan, with the remaining 2% being inherited from Neanderthals through admixture during the Out-of-Africa. I would suggest replacing « up to 5% » with « an additional 3% ».

2/ In Figure 1B, the authors show the number of Vindija/Denisova introgressed alleles falling in open/active chromatin regions and /or disrupt a transcription factor binding site. To my surprise, however, I did not see a breakdown of the percentage of active/differentially active CRS according to the SNP annotation (open/active chromatin & TF disruption. This should be easy to assess and would be very useful for the design of future MPRA experiments. Could the authors provide plots similar to Fig S8A and S8B, but grouping SNPs with the various combinations from Fig 1B?

3/ In figure S1 & S2, the total number of SNPs does not seem to match across panels (eg. between B cells and LCLs). Is there an explanation for this? Or is this an error?

4/ In figure S4, could the authors clarify how the number of barcodes per oligo is computed. Is it the total number of barcodes associated with a specific oligo at the association-sequencing step? Or the number of barcodes present in the plasmid-pool and with non-zero RNA reads in at least one replicate? if the same barcode is present in multiple replicates, is it counted twice, or only unique barcodes are counted?

5/ In figure S7, the correlation between replicates appears rather low, which I suspect is due to sequences with a low number of associated barcodes driving the correlation downwards. Could the authors repeat the analysis from figure S7A, stratifying by number of barcodes.

Reviewer #2: Here, Comerford and co-authors present results from using a massively parallel reporter assay to evaluate regulatory activity across a panel of nearly 26k variants with alleles from different introgression sources (Neanderthal, Denisovan) and states (introgressed, modern) in individuals of Papuan ancestry. This library is tested in several lymphoblastoid lines of different ancestries, and the authors find that in each of the Denisovan and Neanderthal sequence sets, around 8-9% display enhancer activity. Within these active sequences, ~9% have differential activity across modern and archaic alleles.

Overall, I’m enthusiastic about the motivation of the study and results presented here; however, there are several points that could be addressed to enhance the study’s impact. I will note that I have expertise in functional genomics/MAVEs and not in evolutionary biology, so my comments will mostly focus on the former.

Major:

1. I appreciate the use of multiple LCLs representing multiple ancestries. I was wondering why you used two replicates per cell line versus >2 replicates in one or both of the Papuan lines. I understand the value of looking across ancestries, but then that feature of the experimental design is never really commented on or addressed in the results section.

2. You mention the relatively low pairwise correlation of activity you see between replicates which improves when you filter to your positive control set. Have you also looked at pairwise correlation between pDNA replicates? My expectation is that it should be pretty high given that you’re delivering the same library to all samples.

3. You include a set of 300 negative control sequences in your library, but then use the mean cDNA/pDNA counts within a replicate as your threshold for calling sequences active/inactive. Did you consider using your negative controls to estimate a null distribution for this testing instead?

4. Could you clarify your rationale for using barcode-level estimates of activity for statistical testing within each replicate? Because your replicate correlations were somewhat low, I’m not sure that this approach adequately captures this variability.

5. As a related question: with the exception of your positive control sets, the two GM LCL replicates have a consistently higher proportion of uniquely active oligos compared to the other two lines (Supplementary Figure 6). I assume this is just a thresholding effect where marginal cases aren’t captured, but I wasn’t sure if there was some other interpretation.

6. More clarity on how you performed the differential activity testing would be helpful. To identify active oligo sets, you’re performing statistical testing within each replicate and then defining a consensus set across replicates (FDR < 0.05 in at least two). Are you using data across all 5 replicates to perform differential activity testing? You include the model you pass to mpralm, but it would also be helpful to know what the design vector (i.e., other covariates) looks like.

7. You used regulatory annotations to help select the variants tested in the library. You mention that the regulatory annotations are defined in at least one immune cell type – I assume this means that some of your annotations are unique to one or more immune cell types while others are shared across lots of cell types. Are there any worthwhile insights to be gained if you partition the regulatory annotations based on whether they’re unique to immune cells versus shared broadly across a lot of cell types?

8. Your transcription factor binding site results are really exciting! Are you restricting your analysis to only TFs that are expressed in LCLs/immune cells broadly, or any TFs?

9. In your haplotypic analysis, you see that simply summing the effects of individual alleles doesn’t fully capture the activity of the full haplotypes. I am very excited by this subset of the library and think this is sorely lacking from a lot of MPRA studies. It would enhance the impact of this section if you could elaborate on these results a bit more – e.g., do you find any examples where there are clear driver variant(s) that seem to drive the synergistic effects? Is there anything interesting/unique about variants that fall off the diagonal in Figure 3C (top left, bottom right quadrants)?

Minor:

1. Can you add titles to Figure 1B to make it very explicit what purple and orange are since it’s the first time you use this color scheme? It’s nicely labeled elsewhere.

2. Line 258: what percentage of all haplotype-like sequences that were designed does this represent?

3. I appreciate the insights presented in Figure 4H, but I find this figure a little bit hard to follow. I generally read figures from top-to-bottom, so I think it could be helpful to flip the order of results presented on the y-axis which also aligns with how you talk about these data in the results section: first single-SNP oligos, then haplotype-like oligos.

4. The observations presented in lines 469-473 are striking/disappointing, and really point to the importance of the work you’re presenting here. I completely agree with your claims in the final lines of the discussion (“MPRAs and other modalities of multiplex assays of variant effects (MAVEs)...” I’d recommend reiterating your findings in lines 294-297 to provide further validity – it’s not just about the feasibility of MAVEs versus generating large-scale datasets in non-Europeans, but also that they’re quite accurate!

**Have all data underlying the figures and results presented in the manuscript been provided?**

Large-scale datasets should be made available via a public repository as described in the *PLOS Genetics*
data availability policy , and numerical data that underlies graphs or summary statistics should be provided in spreadsheet form as supporting information., and numerical data that underlies graphs or summary statistics should be provided in spreadsheet form as supporting information.

Reviewer #1: Yes

Reviewer #2: Yes

PLOS authors have the option to publish the peer review history of their article (what does this mean? . If published, this will include your full peer review and any attached files.). If published, this will include your full peer review and any attached files.

**Do you want your identity to be public for this peer review?** For information about this choice, including consent withdrawal, please see our For information about this choice, including consent withdrawal, please see our Privacy Policy ..

Reviewer #1: **Yes:** Maxime RotivalMaxime Rotival

Reviewer #2: No

**Figure resubmission:**
---

## [Decision Letter · Decision Letter 1]

24 Nov 2025

PGENETICS-D-25-00637R1

Mapping the gene regulatory landscape of archaic hominin introgression in modern Papuans

PLOS Genetics

Dear Dr. Gallego Romero,

Thank you for submitting your manuscript to PLOS Genetics. After careful consideration, we feel that it has merit but does not fully meet PLOS Genetics's publication criteria as it currently stands. Therefore, we invite you to submit a revised version of the manuscript that addresses the remaining points raised by one of the reviewers.

Please submit your revised manuscript within by Dec 24 2025 11:59PM. If you will need more time than this to complete your revisions, please reply to this message or contact the journal office at plosgenetics@plos.org. Please include the following items when submitting your revised manuscript:

* A rebuttal letter that responds to each point raised by the reviewer. You should upload this letter as a separate file labeled 'Response to Reviewers'. This file does not need to include responses to formatting updates and technical items listed in the 'Journal Requirements' section below.

We look forward to receiving your revised manuscript.

Kind regards,

Francesca Luca

Guest Editor

PLOS Genetics

Justin Fay

Section Editor

PLOS Genetics

Aimée Dudley

Editor-in-Chief

PLOS Genetics

Anne Goriely

Editor-in-Chief

PLOS Genetics

**Reviewers' comments:**

Reviewer's Responses to Questions

**Comments to the Authors:**

Reviewer #1: The authors have addressed all my remarks adequately, and have significantly strengthened their statistical analyses and the manuscript as a whole.

My only remaining concern relates to the updated figure 3, and the associated statements. « We were able to identify haplotypes for which all alleles and corresponding single variant sequences were active, but were the combination of allele showed greater activity than the single variant sequence alone (Figure 3B), suggesting addivity »

If I understand figure 3B correctly, the authors have actually tested 8 sequences for the pair of variants located at chr6 :107,746,260 & chr6 :107,746,429 ie. 169 bp away from each other.

- Two 170 bp sequences centered on 107,746,260 : (i.e. from ~107,746,175 to 107,746,345 bp), called SNP1-centered sequences 1H & 1D on the figure 3B

- Two 170 bp sequences centered on 107,746,429 : (i.e. from ~107,746,344 to 107,746,514 bp), called SNP2-centered sequences 2H & 2D on the figure 3B

- Four 170bp sequences containing each combination of alleles from both variants: (i.e. from ~107,746,260 to 107,746,430 bp) callled haplotypic sequences HH, HD, DH, and DD on the figure 3B.

If that’s correct, the authors can/should actually answers two separate questions :

1- Is the effect of each variant dependent on its position on the tested sequence ?

a. is (1D-1H), the effect of SNP1 positionned at the center of the tested sequence the same as (DH-HH) the effect of SNP1 positionned at the border of the tested (human) sequence ?

b. is (2D-2H), the effect of SNP2 positionned at the center of the tested sequence the same as (HD-HH) the effect of SNP2 positionned at the border of the tested (human) sequence ?

2- Do archaic variants exhibit non-additive effect on gene expression ?

a. is (DH-HH) the effect of SNP1 on a human background, the same as (DD-HD), the effect of the effect of SNP1 on an archaic background ?

b. is (HD-HH) the effect of SNP2 on a human background, the same as (DD-DH), the effect of the effect of SNP2 on an archaic background ?

What Is see from figure 3B would suggest that in the haplotypic sequences, SNP1 and SNP2 act in an additive manner, with SNP1 decreasing expression and SNP2 increasing expression, but the effect of SNP2 is position-dependent and is not detected in the SNP2-centered comparison (proper statistical testing for differntial activity would however be in order here).

I think however that stating that « the combination of allele showed greater activity than the single variant sequence » is misleading, as a suggests a GxG interaction which is not supported by the data in this case.

Overall, given the comment to reviewer 2 regarding the difficulty to interpret off diagonal points in Figure 3G, I wonder if it would not be more staightforward to focus on haplotypic constructs, and compare for each variant, the effect on the human background vs effect on the archaic background (Question 1). A comparison of SNP effects between SNP-centrered or (human) haplotypic contructs (Question 2) could also be shown as a word of caution on how the design of the sequence construct may affect the inferred effect of tested variants.

finally, it would be nice to highlight in figures 3F, 3G and S17 the specific SNPS that the authors choose to highlight in the Figure 3B-3E.

Reviewer #2: The authors' revisions have satisfied all of my previous concerns with the manuscript.

**Have all data underlying the figures and results presented in the manuscript been provided?**

Large-scale datasets should be made available via a public repository as described in the *PLOS Genetics*
data availability policy , and numerical data that underlies graphs or summary statistics should be provided in spreadsheet form as supporting information., and numerical data that underlies graphs or summary statistics should be provided in spreadsheet form as supporting information.

Reviewer #1: Yes

Reviewer #2: Yes

PLOS authors have the option to publish the peer review history of their article (what does this mean? . If published, this will include your full peer review and any attached files.). If published, this will include your full peer review and any attached files.

**Do you want your identity to be public for this peer review?** For information about this choice, including consent withdrawal, please see our For information about this choice, including consent withdrawal, please see our Privacy Policy ..

Reviewer #1: **Yes:** Maxime ROTIVALMaxime ROTIVAL

Reviewer #2: No

**Figure resubmission:**
---

## [Decision Letter · Decision Letter 2]

3 Feb 2026

PGENETICS-D-25-00637R2

Mapping the gene regulatory landscape of archaic hominin introgression in modern Papuans

PLOS Genetics

Dear Dr. Gallego Romero,

Thank you for submitting your manuscript to PLOS Genetics. After careful consideration, we feel that it has merit but does not fully meet PLOS Genetics's publication criteria as it currently stands. Therefore, we invite you to submit a revised version of the manuscript that addresses the minor points raised during the review process.

Please submit your revised manuscript within by Mar 05 2026 11:59PM. If you will need more time than this to complete your revisions, please reply to this message or contact the journal office at plosgenetics@plos.org. Please include the following items when submitting your revised manuscript:

We look forward to receiving your revised manuscript.

Kind regards,

Francesca Luca

Guest Editor

PLOS Genetics

Justin Fay

Section Editor

PLOS Genetics

Aimée Dudley

Editor-in-Chief

PLOS Genetics

Anne Goriely

Editor-in-Chief

PLOS Genetics

**Additional Editor Comments:**

Please address the reviewer's latest comment either by performing the suggested analyses or modifying the text of the current results to more clearly state the scope of the analyses already performed and to clarify their interpretation.

**Journal Requirements:**

**Reviewers' comments:**

Reviewer's Responses to Questions

**Comments to the Authors:**

Reviewer #1: I thank the authors for addressing my remarks and clarifying some of their analyses.

I do feel however that the authors have slightly missed my point regarding question 2 and the assessment of GxG interactions.

In their answer l 304-312, the authors assess the rate of concordance in activity between DD and DH and between DD and HD, see that these rates are similar, and conclude that activity is robust to the haplotype sequence context. In my opinion, this shows that the rate of emVars is independent of localization on the haplotype (which is expected), but doesn’t test robustness of SNP effect to sequence context.

I think that the most relevant is to quantify how often :

- activity of DD and DH are discordant AND activity of HD and HH are concordant.

OR

- activity of DD and DH are concordant AND activity of HD and HH are discordant.

OR

- activity of DD and HD are discordant AND activity of DH and HH are concordant.

OR

- activity of DD and HD are concordant AND activity of DH and HH are discordant.

Could the authors quantify this instead ?

Indeed, any of these four cases would be indicative of strong GxG interaction.

The lack of such cases also would be worth reporting as it would emphasize the lack of strong GxG interactions. The authors could then conclude that DIFFERENTIAL activity between alleles is generally robust to the haplotype sequence context.

Regarding the first point, I do feel that a discordance of 12%-15% in allelic activity depending on SNP position, is actually pretty high and highlights the importance of focusing on differential activity between alleles (ie. emVars) rather than absolute activity…. This is more a matter of appreciation however, and I have no issue with the current, more optimistic, formulation of the results (l286-l303).

**Have all data underlying the figures and results presented in the manuscript been provided?**

Large-scale datasets should be made available via a public repository as described in the *PLOS Genetics*
data availability policy , and numerical data that underlies graphs or summary statistics should be provided in spreadsheet form as supporting information., and numerical data that underlies graphs or summary statistics should be provided in spreadsheet form as supporting information.

Reviewer #1: Yes

PLOS authors have the option to publish the peer review history of their article (what does this mean? . If published, this will include your full peer review and any attached files.). If published, this will include your full peer review and any attached files.

**Do you want your identity to be public for this peer review?** For information about this choice, including consent withdrawal, please see our For information about this choice, including consent withdrawal, please see our Privacy Policy ..

Reviewer #1: **Yes:** Maxime ROTIVALMaxime ROTIVAL

**Figure resubmission:**
---

## [Editor Report · Decision Letter 3]

21 Feb 2026

Dear Dr Gallego Romero,

We are pleased to inform you that your manuscript entitled "Mapping the gene regulatory landscape of archaic hominin introgression in modern Papuans" has been editorially accepted for publication in PLOS Genetics. Congratulations!

Yours sincerely,

Francesca Luca

Guest Editor

PLOS Genetics

Justin Fay

Section Editor

PLOS Genetics

Aimée Dudley

Editor-in-Chief

PLOS Genetics

Anne Goriely

Editor-in-Chief

PLOS Genetics

BlueSky: @plos.bsky.social

Comments from the reviewers (if applicable):

**Data Deposition**

If you have submitted a Research Article or Front Matter that has associated data that are not suitable for deposition in a subject-specific public repository (such as GenBank or ArrayExpress), one way to make that data available is to deposit it in the Dryad Digital Repository . As you may recall, we ask all authors to agree to make data available; this is one way to achieve that. A full list of recommended repositories can be found on our . As you may recall, we ask all authors to agree to make data available; this is one way to achieve that. A full list of recommended repositories can be found on our website ..

http://datadryad.org/submit?journalID=pgenetics&manu=PGENETICS-D-25-00637R3

Additionally, please be aware that our data availability policy  requires that all numerical data underlying display items are included with the submission, and you will need to provide this before we can formally accept your manuscript, if not already present. requires that all numerical data underlying display items are included with the submission, and you will need to provide this before we can formally accept your manuscript, if not already present.

**Press Queries**

If you or your institution will be preparing press materials for this manuscript, or if you need to know your paper's publication date for media purposes, please inform the journal staff as soon as possible so that your submission can be scheduled accordingly. Your manuscript will remain under a strict press embargo until the publication date and time. This means an early version of your manuscript will not be published ahead of your final version. PLOS Genetics may also choose to issue a press release for your article. If there's anything the journal should know or you'd like more information, please get in touch via plosgenetics@plos.org ..

---

## [Editor Report · Acceptance letter]

PGENETICS-D-25-00637R3

Mapping the gene regulatory landscape of archaic hominin introgression in modern Papuans

Dear Dr Gallego Romero,

We are pleased to inform you that your manuscript entitled "Mapping the gene regulatory landscape of archaic hominin introgression in modern Papuans" has been formally accepted for publication in PLOS Genetics! Your manuscript is now with our production department and you will be notified of the publication date in due course.

With kind regards,

Zsofia Freund

PLOS Genetics

On behalf of:
